# Distinct representations of body and head motion are dynamically encoded by Purkinje cell populations in the macaque cerebellum

**Omid A Zobeiri[1], Kathleen E Cullen[2,3,4,5]\***

[1]Department of Biomedical Engineering, McGill University, Montreal, Canada; [2]Department of Biomedical Engineering, Johns Hopkins University, Baltimore, United States; [3]Department of Otolaryngology-Head and Neck Surgery, Johns Hopkins University School of Medicine, Baltimore, United States; [4]Department of Neuroscience, Johns Hopkins University School of Medicine, Baltimore, United States; [5]Kavli Neuroscience Discovery Institute, Johns Hopkins University, Baltimore, United States

**\*For correspondence:**
kathleen.cullen@jhu.edu

**Competing interest:** The authors declare that no competing interests exist.

**Abstract** The ability to accurately control our posture and perceive our spatial orientation during self-motion requires knowledge of the motion of both the head and body. However, while the vestibular sensors and nuclei directly encode head motion, no sensors directly encode body motion. Instead, the integration of vestibular and neck proprioceptive inputs is necessary to transform vestibular information into the body-centric reference frame required for postural control. The anterior vermis of the cerebellum is thought to play a key role in this transformation, yet how its Purkinje cells transform multiple streams of sensory information into an estimate of body motion remains unknown. Here, we recorded the activity of individual anterior vermis Purkinje cells in alert monkeys during passively applied whole-body, body-under-head, and head-on-body rotations. Most Purkinje cells dynamically encoded an intermediate representation of self-motion between head and body motion. Notably, Purkinje cells responded to both vestibular and neck proprioceptive stimulation with considerable heterogeneity in their response dynamics. Furthermore, their vestibular responses were tuned to head-on-body position. In contrast, targeted neurons in the deep cerebellar nuclei are known to unambiguously encode either head or body motion across conditions. Using a simple population model, we established that combining responses of ~40-50 Purkinje cells could explain the responses of these deep cerebellar nuclei neurons across all self-motion conditions. We propose that the observed heterogeneity in Purkinje cell response dynamics underlies the cerebellum's capacity to compute the dynamic representation of body motion required to ensure accurate postural control and perceptual stability in our daily lives.

## Editor's evaluation

This paper addresses the important question of how the cerebellum transforms multiple streams of sensory information into an estimate of the motion of the body in the world. The authors find that Purkinje cells, the inhibitory principal neurons of the cerebellar cortex, have multimodal and highly diverse responses to vestibular and neck proprioceptive inputs. Notably, this information is combined in a way that is different from what is seen in downstream fastigial neurons, which reflect either head or body motion, but not both.

## Introduction

The cerebellum guides motor performance by computing differences between the expected versus actual consequences of movements and then adjusting the commands sent to the motor system (reviewed in *Wolpert et al., 1998*; *Raymond and Medina, 2018*). Patients with damage to the anterior vermis of the cerebellum show impaired posture and balance, as well as deficits in motor coordination (*Diener et al., 1984*; *Bastian et al., 1998*; *Ilg et al., 2008*; *Sullivan et al., 2006*; *Mitoma et al., 2020*). In this context, the anterior vermis has a vital role in the vestibulospinal pathways that generate the postural adjustments required to ensure the maintenance of balance during our everyday activities. Additionally, there is an emerging consensus that the cerebellum contributes to our self-motion perception. Indeed, patients with degeneration of the cerebellar vermis demonstrate reduced perceptual time constants and detection thresholds to externally applied rotations (*Bronstein et al., 2008*; *Dahlem et al., 2016*).

The prevailing view is that the vestibular pathways mediating vestibulospinal reflexes and the stable perception of self-motion explicitly transform vestibular information from a head-centered to a body-centered reference frame. The vestibular sensory organs are located within the head, making the vestibular system's native reference frame head-centered (reviewed in *Cullen, 2019*). In turn, vestibular nerve afferents and their targets in the vestibular nuclei also encode information in a head-centered reference frame (*Roy and Cullen, 1998*; *Roy and Cullen, 2001*; *Roy and Cullen, 2004*; *Carriot et al., 2013*; *Brooks and Cullen, 2014*; *Sadeghi et al., 2007*; *Jamali et al., 2009*; *Cullen and Minor, 2002*). However, the brain accounts for the position of the head relative to the body for vestibulospinal reflexes to accurately control the musculature required to maintain upright posture and balance (*Tokita et al., 1989*; *Tokita et al., 2009 Kennedy and Inglis, 2002*). Indeed, distinct representations of body versus head motion are encoded by individual neurons in the fastigial nucleus (*Brooks and Cullen, 2009*; *Brooks and Cullen, 2014*) – the most medial of deep cerebellar nuclei – which lesion studies have shown serves an important role in the control of posture and balance (*Thach et al., 1992*; *Kurzan et al., 1993*; *Pélisson et al., 1998*). Neck proprioceptors provide the head position information required for this transformation (reviewed in *Cullen and Zobeiri, 2021*). Thus, the integration of neck proprioceptive and vestibular signals is thought to underlie the transformation from a head-centered to a body-centered reference frame in vestibulospinal reflexes pathways as well as our ability to perceive body motion independently of head motion (*Mergner et al., 1997*; *Peterka, 2002*).

There are many reasons to believe that the anterior region of the cerebellar vermis is vital in the transformation of vestibular information from a head-centered to a body-centered reference frame. First, Purkinje cells in this region project to the rostral portion of the fastigial nucleus (*Batton et al., 1977*; *Yamada and Noda, 1987*), which lesion studies have shown serves an important role in the control of posture and balance (*Thach et al., 1992*; *Kurzan et al., 1993*; *Pélisson et al., 1998*). Second, inhibition of the cerebellar vermis via continuous theta-burst stimulation impairs the modulation of vestibulospinal pathways that normally accounts for changes in the position of the head relative to the body (*Lam et al., 2016*). Third, neuronal recordings from anterior vermis Purkinje cells in decerebrate cats have demonstrated that individual neurons can encode both vestibular and neck proprioceptive-related information (*Denoth et al., 1979*; *Manzoni et al., 1998*; *Manzoni et al., 1999*; *Manzoni et al., 2004*), thereby providing a neural substrate for the coordinate transformation using neck proprioceptive signals to convert head-centered vestibular signals to a body-centered reference frame. However, these studies stopped short of establishing whether Purkinje cells integrate vestibular and neck proprioceptive signals to dynamically encode head or body movement.

Thus, a key question yet to be answered is: Does the cerebellum integrate vestibular and neck proprioceptive signals to provide a dynamic representation of body motion relative to space? Here, we recorded the activity of single Purkinje cells in the anterior vermis during head motion, body motion, and combined head and body motion. We found considerable heterogeneity across individual Purkinje cells in their encoding of head versus body motion, with most (~75%) neurons dynamically encoding an intermediate representation of self-motion between head and body motion. These neurons, termed bimodal neurons, responded to both vestibular and neck proprioceptive stimulation and displayed head-position-dependent tuning in their sensitivity to vestibular stimulation. In contrast, a minority of cells, termed unimodal neurons, only responded to vestibular stimulation and unambiguously encoded the motion of the head in space. Across all cells, the linear combination of

a given neuron's response sensitivity to dynamic neck and vestibular stimulation alone well estimated its response during combined stimulation. Finally, we found that a simple linear combination model combining the responses of ~40 Purkinje cells could account for the more homogeneous responses of target neurons in the deep cerebellar nuclei (i.e., the rostral fastigial nucleus [rFN]) during applied self-motion. Our results provide the first evidence, at the level of single Purkinje cells, that a sequential transformation from a head-centered to body-centered reference frame occurs between the cerebellum and deep cerebellar nucleus to ensure postural and perceptual stability in everyday life.

## Results

### Most vestibular-sensitive Purkinje cells in the anterior vermis are also sensitive to stimulation of neck proprioceptors

Each Purkinje cell in our population ($n$ = 73) was responsive to vestibular stimulation and was insensitive to eye movements. To assess each neuron's vestibular sensitivity, we applied ipsilaterally and contralaterally directed whole-body rotations in the dark (i.e., whole-body-rotations; see Materials and methods). As illustrated in *Figure 1A*, we found considerable heterogeneity in vestibular sensitivities across our Purkinje cell population. Some neurons generated excitatory versus inhibitory responses for oppositely directed head movements (*Figure 1A*; left, linear). Alternately, some neurons generated bidirectional excitatory responses (center, v-shaped), while others largely only generated excitatory responses for one movement direction (*Figure 1A*; right, rectifying). This contrasts with the vestibular responses recorded in areas targeted by Purkinje cells in the anterior vermis. Notably, vestibular-only neurons in the rFN and vestibular nucleus consistently show excitatory versus inhibitory responses for oppositely directed head movements (rFN: *Gardner and Fuchs, 1975*; *Shaikh et al., 2005*; vestibular nucleus: *Scudder and Fuchs, 1992*; *Cullen and McCrea, 1993*; *McCrea et al., 1999*; *Roy and Cullen, 2004*).

To quantify the vestibular sensitivity of each Purkinje cell in our population, we fit a least-squares dynamic regression model with three kinematic terms (i.e., head-in-space position, velocity, and acceleration) to responses for movements in each direction (*Figure 1—figure supplement 1*, see Materials and methods). We found that the preferred movement direction (i.e., the direction that resulted in an excitatory response, or in the greater excitatory response in the case of v-shaped neurons) could be either ipsilateral or contralateral for a given Purkinje cell. Neurons with preferred responses for ipsilaterally ($n$ = 32, e.g., *Figure 1A*, right panel) or contralaterally ($n$ = 41, *Figure 1A*, left and middle panels) directed rotations were accordingly classified as Type I or Type II, respectively. Our analysis further revealed that the response dynamics varied considerably across neurons, with 43% neurons demonstrating responses that were relatively in-phase (±15°) with head velocity (1.8 ± 7.7°), while others demonstrated marked response leads (32%, 57 ± 31) or lags (25%, –47.5 ± 22°). *Figure 1B, C* illustrates the vestibular response vectors for our populations of neurons computed in their preferred and non-preferred directions, respectively. The vector represents the gain (length) and phase (angle) of the neural responses to each stimulus computed at 1 Hz (see Materials and methods). The large arrows represent average neuronal responses to vestibular stimuli for Type I ($S_{vest.}$ = 0.42 ± 0.37 (sp/s)/(°/s), $Phase_{vest.}$ = 6 ± 31) and Type II neurons ($S_{vest.}$ = 0.31 ± 0.34 (sp/s)/(°/s), $Phase_{vest.}$ = 172 ± 42°), respectively.

We next addressed whether the Purkinje cells that responded to vestibular stimulation also responded to the activation of neck proprioceptors. To assess each neuron's proprioceptive sensitivity, we applied ipsilaterally and contralaterally directed rotations to the monkey's body while its head was held stationary relative to space (i.e., body-under-head rotations; see Materials and methods), with the same motion profiles as those used for the assessment of vestibular sensitivities. Thus, since the head did not move relative to space, neck proprioceptors but not the vestibular system were stimulated in this condition. *Figure 2A* illustrates the responses recorded from the same three example neurons shown in *Figure 1A*. We quantified the neck proprioceptive sensitivity of each Purkinje cell using least-squares dynamic regression (see Materials and methods) and found that most neurons (~75%, $n$ = 54) were sensitive to passive proprioceptive stimulation (*Figure 2B, C*, filled bars; bimodal neurons), whereas the remaining ~25% ($n$ = 19) were insensitive (*Figure 2*, *Figure 2B, C*, open bars; unimodal neurons). Overall, similar to our findings above regarding vestibular stimulation, the dynamics of responses to proprioceptive stimulation varied considerably across Purkinje cells (*Figure 2—figure*

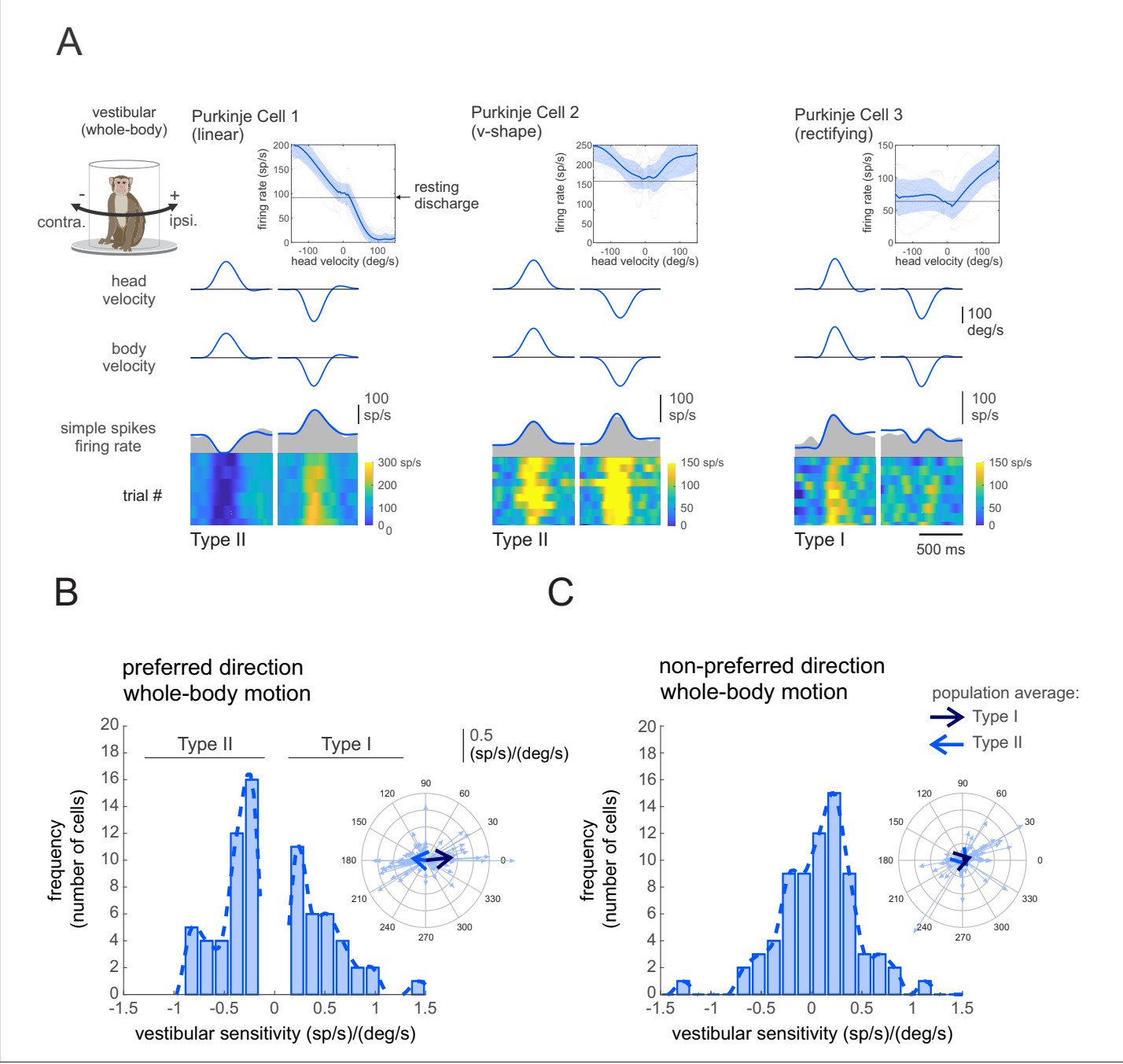

**Figure 1.** Purkinje cell simple spike's responses to vestibular stimulation. (**A**) Vestibular stimulation was generated by applying passive whole-body rotations about the vertical axis. The resulting neural responses are shown for three example Purkinje cells. The top two rows illustrate rotational head and body velocities. The bottom row shows the simple spike firing rate (gray shaded regions) with the linear estimation of the firing rate based on head motion superimposed (blue traces). The heat maps show the simple spike firing rate for individual trials. *Insets*: the relationship between simple spike firing rate (phase-corrected) and angular head-in-space velocity. (**B,C**) Distribution of vestibular sensitivities for motion in the preferred (**B**) direction (i.e., the direction resulting in the larger increase in simple spike firing rate) and non-preferred (**C**) direction. The dashed lines are fits on the distributions. Note, by convention positive and negative values in (**B**) represent cells with Type I versus II vestibular responses (i.e., preferred direction was ipsilateral versus contralateral, respectively). *Insets*: polar plots where the vector length and angle represent each neuron's vestibular response sensitivity and phase, respectively. Filled and open arrows represent the population-averaged vectors for Type I and II cells, respectively.

The online version of this article includes the following figure supplement(s) for figure 1:

**Figure supplement 1.** Purkinje cells show heterogeneity in their simple spike responses to vestibular stimulation.

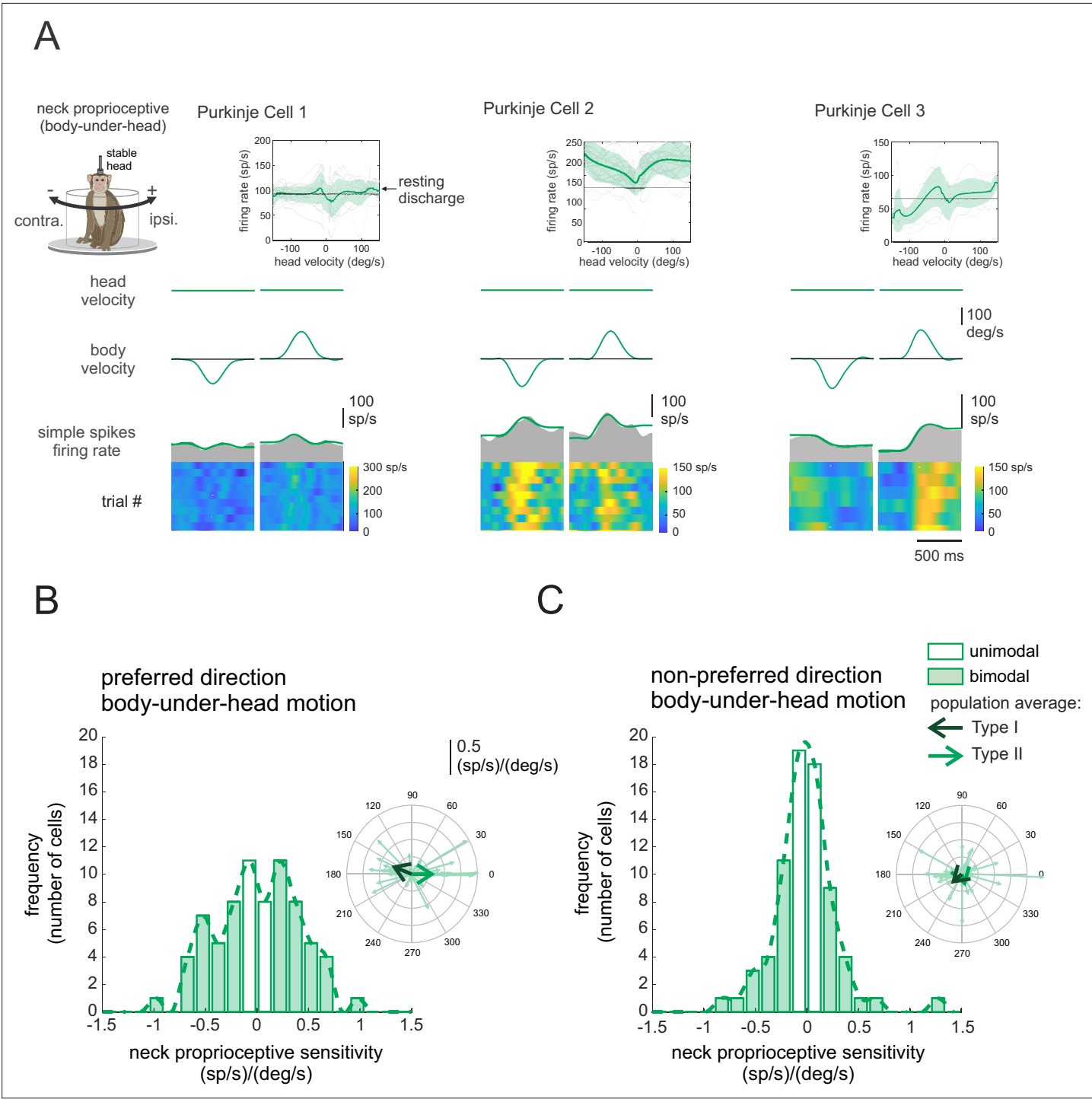

**Figure 2.** Purkinje cell simple spike's responses to neck proprioceptive stimulation. (**A**) Proprioceptive stimulation was generated by applying body-under-head rotation about the vertical axis while holding the head earth. The resulting neural responses are illustrated for the same three example Purkinje cells shown above in *Figure 1*. The top two rows illustrate rotational head and body velocities. The bottom row shows the resultant simple spike firing rate (gray shaded regions) with the linear estimation of the firing rate based on body motion superimposed (green traces). The heat maps show the simple spike firing rate for individual trials. *Insets*: the relationship between simple spike firing rate (phase-corrected) and angular body-in-space velocity. (**B, C**) Distribution of proprioceptive sensitivities for the preferred (**B**) and non-preferred (**C**) directions of body movement. Filled versus open bars represent neurons that were sensitive versus insensitive to neck proprioceptive stimulation (i.e., bimodal versus unimodal cells, respectively). The dashed lines are fits on the distributions. *Insets*: polar plots where the vector length and angle represent each neuron's proprioceptive response sensitivity and phase, respectively. Filled versus open arrows represent the population-averaged vectors for neurons with Type I versus II vestibular responses (i.e., *Figure 1*), respectively.

*Figure 2 continued on next page*

*Figure 2 continued*

The online version of this article includes the following figure supplement(s) for figure 2:

**Figure supplement 1.** Purkinje cells show heterogeneity in their simple spike responses to proprioceptive stimulation.

**Figure supplement 2.** Purkinje cells show heterogeneity in their simple spike responses to vestibular versus proprioceptive stimulation.

**Figure supplement 3.** For most of the Purkinje cells, the responses to vestibular and neck proprioceptive stimulation were classified in different groups.

*supplement 1*). The insets in *Figure 2B, C* illustrate the response vectors for proprioceptive stimulation across our populations of neurons in the preferred and non-preferred directions, respectively. As in *Figure 1B, C* above, the vector was computed based on the gain (length) and phase (angle) of the neural responses to each stimulus at 1 Hz (see Materials and methods) and large arrows represent average neuronal responses measured in response to proprioceptive stimuli for neurons with Type I ($S_{prop.}$ = 0.12 ± 0.44 (sp/s)/(°/s), $Phase_{prop.}$ = 159 ± 29°) versus Type II ($S_{prop.}$ = 0.13 ± 0.46 (sp/s)/(°/s), $Phase_{prop.}$ = –27 ± 20°) responses. Furthermore, Purkinje cells showed considerable heterogeneity in their simple spike response dynamics to vestibular versus proprioceptive stimulation (*Figure 2— figure supplement 2*). Indeed, neurons typically did not show the same patterns (linear, v-shaped, rectified) to vestibular versus proprioceptive stimulation (*Figure 2—figure supplement 3*).

## Purkinje cell's responses to simultaneous proprioceptive and vestibular stimulation

So far, we have shown that most Purkinje cells in our population were sensitive to neck proprioceptive as well as vestibular stimulation, and that we categorized these neurons as 'bimodal' versus neurons that were only responsive to vestibular stimulation as 'unimodal'. The vestibular sensitivities of bimodal Purkinje cells were comparable to those of their unimodal counterparts (p = 0.17). We further found that the neck sensitivities of bimodal neurons were most often (67%) antagonistic relative to their vestibular sensitivities in the *preferred* direction. This can be seen in *Figure 3A and B*, where the average vectors representing the neuronal response to the vestibular and proprioceptive stimulation point in opposite directions (*Figure 3A and B*, thick blue versus green arrows, respectively). In contrast, relative to vestibular sensitivities in *non-preferred* direction, bimodal cells were as likely to have antagonistic as agonistic responses to neck proprioceptive stimulation (*Figure 3— figure supplement 1*).

During everyday activities, we move our head relative to our body, and thus simultaneously activate both vestibular sensors and neck proprioceptors. To directly establish how vestibular and neck proprioceptive information is integrated in the anterior vermis, we next recorded the responses of the same Purkinje cell populations during combined stimulation of neck proprioceptors and the vestibular system. Specifically, we applied ipsilaterally and contralaterally directed rotations of the monkey's head relative to its earth-stationary body (i.e., head-on-body rotations; see Materials and methods), again with the same motion profiles as those used above in the assessment of neuronal vestibular and neck proprioceptive sensitivities. The responses of the same three example neurons above in *Figures 1 and 2* are shown in *Figure 3C*. The head motion-based linear estimation of firing rate (solid black traces, see Materials and methods) is plotted on the firing rate for each cell. A firing rate prediction (dashed red traces) based on the linear summation of each neuron's sensitivity to vestibular and neck proprioceptive stimulation when each was applied in isolation (i.e., *Figures 1 and 2*, respectively) is superimposed for comparison. The example neurons were typical in that each neuron's modulation for combined stimulation in the *preferred direction* (i.e., gray columns) was well predicted by the linear summation of the neuron's vestibular and proprioception sensitivities. The polar plots show the vector summation (dashed red arrow) of the example neuron's response to vestibular and proprioceptive stimulation (blue and green arrows) when applied in isolation versus the response vector for combined stimulation (black arrow; *Figure 3—figure supplement 2*). Correspondingly, there was good alignment between the vector length and direction computed for the firing rate estimate and prediction for these three example neurons for the head-on-body motion in the *preferred direction*.

*Figure 3D and E* summarizes the population data for unimodal and bimodal Purkinje cells in each of the three stimulation conditions. Average response sensitivities are shown for *preferred* (*Figure 3D*) and *non-preferred* (*Figure 3E*) direction motion. Note that since there were no significant differences between the response of Type I and II cells (other than their preferred direction), we reported the

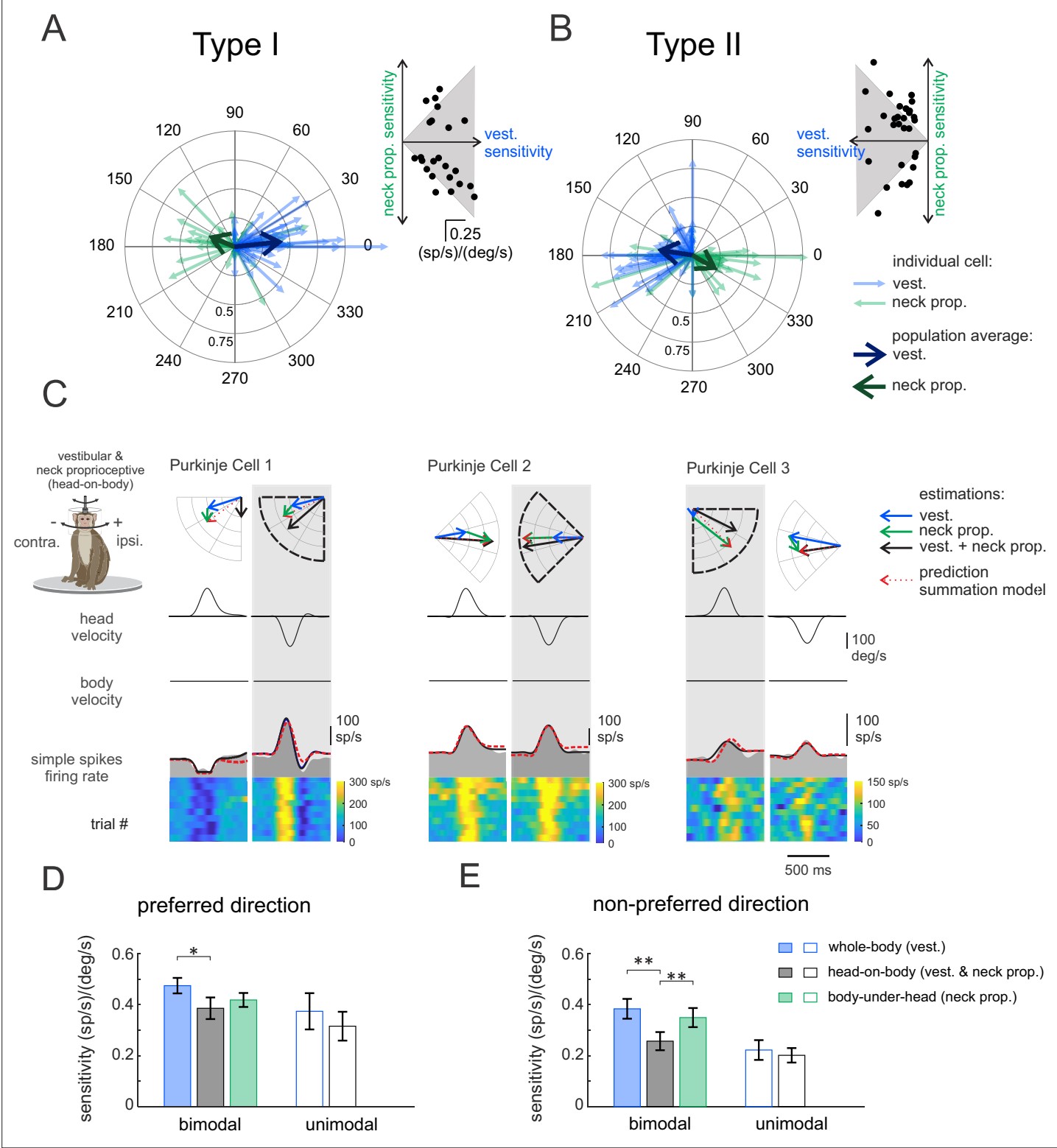

**Figure 3.** Purkinje cells simple spike's responses to combined vestibular-proprioceptive stimulation. (**A, B**) Polar plots illustrating the vestibular (blue) and neck proprioceptive (green) neuronal response sensitivities of Type I (**A**) and Type II (**B**) Purkinje cells for preferred direction of vestibular stimulation and complementary direction proprioceptive stimulation (i.e., body-under-head motion). Bold blue and green arrows represent the mean population vectors, respectively. *Inset*: scatter plots comparing the sensitivity of Type I (**A**) and Type II (**B**) Purkinje cells to vestibular and neck proprioceptive inputs. (**C**) Combined vestibular-proprioceptive stimulation was generated by applying passive head-on-body rotations about the vertical axis. The resulting

*Figure 3 continued on next page*

*Figure 3 continued*

neural responses are shown for the same three example Purkinje cells shown above in *Figures 1 and 2*. The top two rows illustrate rotational head and body velocity. The bottom row shows the resultant simple spike firing rate (gray shaded regions). The linear estimation of firing rate based on head motion (solid black traces) and the firing rate prediction based on the linear summation of neck proprioceptive and vestibular sensitivities (dashed red traces) are both superimposed. Each neuron's preferred motion direction for vestibular stimulation is indicated by the gray column. Polar plots (*top*) represent the sensitivity and phase of each neuron's response to vestibular, proprioceptive, and combined stimulation as well as the response predicted by the summation model. (**D, E**) Bar plots comparing the sensitivities of bimodal and unimodal Purkinje cells to vestibular, proprioceptive, and combined stimulation in the *preferred* (**D**) and *non-preferred* (**E**) motion directions, as defined by each neuron's responses to vestibular stimulation. The response sensitivities of Type I and II neurons are reported as positive values relative to ipsilaterally and contralaterally directed head movements, respectively, to facilitate comparison across all Purkinje cells.

The online version of this article includes the following figure supplement(s) for figure 3:

**Figure supplement 1.** Purkinje cell's responses to combined vestibular-neck proprioceptive stimulation in the non-preferred direction of vestibular stimulation.

**Figure supplement 2.** Polar representations of Purkinje cells simple spike's response.

responses of both groups together, by accounting for the difference in the direction of the modulation of Type II cells. We first hypothesized that the vestibular sensitivity of unimodal neurons should remain constant across conditions regardless of whether the neck proprioceptors were stimulated. Consistent with this proposal, we found that unimodal cell response sensitivities (open bars) were comparable during the vestibular-only and combined stimulation conditions (*Figure 3D and E*, p > 0.22). Likewise, response phases of unimodal neurons were comparable for both conditions (*preferred*: 11 ± 5 versus 11 ± 14; p = 0.21; versus *non-preferred*: 27 ± 8 versus 14 ± 18; p = 0.77). In contrast, we hypothesized that since the vestibular and proprioceptive sensitivities of bimodal Purkinje cells were generally antagonist (i.e., *Figure 3A*, *Figure 3—figure supplement 1*), the oppositely modulated inputs from neck proprioceptors should effectively suppress the vestibular-driven responses during the combined conditions. Indeed, consistent with this prediction, the sensitivities of bimodal Purkinje

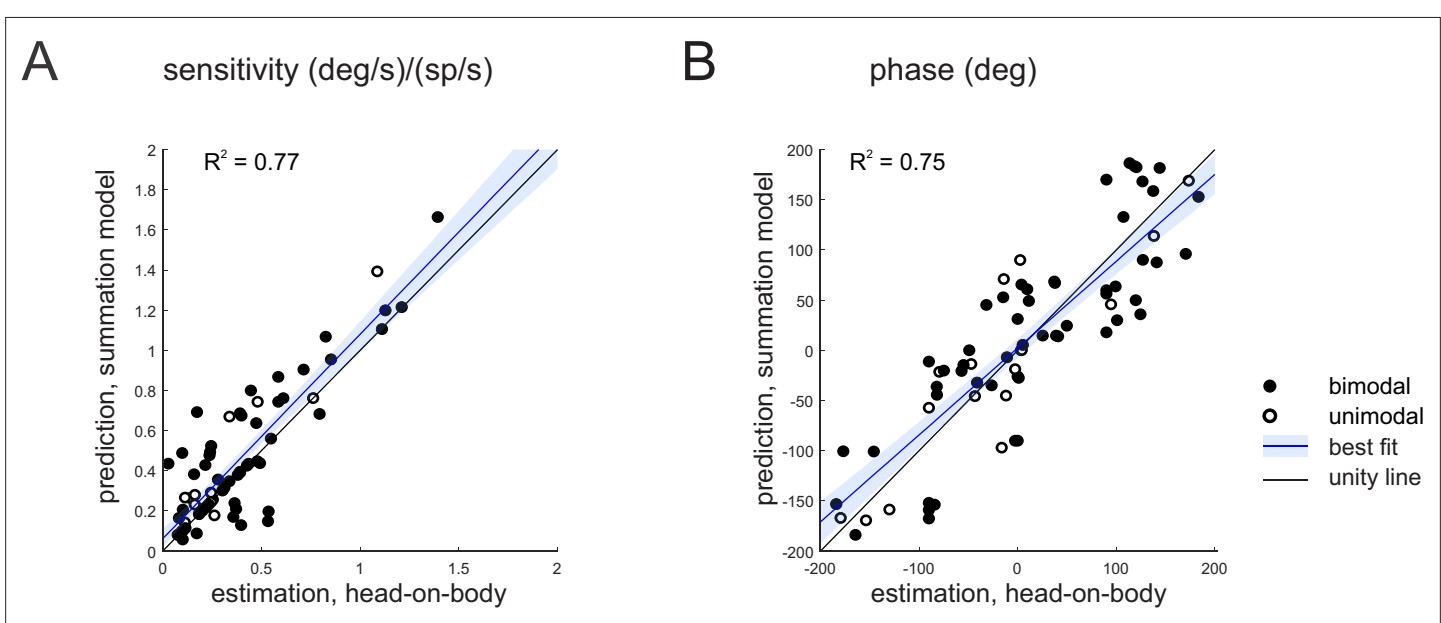

**Figure 4.** Purkinje cell simple spike's responses to combined stimulation are well predicted by the linear summation of a given neuron's responses to vestibular and proprioceptive stimulation when applied alone. (**A, B**) Comparison of estimated and predicted sensitivities (**A**) and phases (**B**) of Purkinje cell's responses to head-on-body rotations in the preferred movement direction. The linear summation of a given neuron's vestibular and neck proprioceptive sensitivities well predicts both sensitivity and phase measures in the combined condition. Blue lines and shading denote the mean ± 95% CI of linear fit.

The online version of this article includes the following figure supplement(s) for figure 4:

**Figure supplement 1.** Purkinje cell simple spike's responses to combined stimulation are well predicted by the linear summation of a given neuron's responses to vestibular and proprioceptive stimulation when applied alone.

cells were reduced during the combined stimulation condition relative to the vestibular-only condition (*Figure 3D and E*, filled bars, p < 0.027).

Above, we showed that the responses of our three example neurons in the combined stimulation were well predicted by the linear summation of the neuron's vestibular and proprioception sensitivities applied in isolation, particularly for stimulation in the *preferred direction*. We next explicitly addressed whether this simple linear model of vestibular-proprioceptive integration could reliably predict neuronal responses in the combined condition across our population of Purkinje cells. To do this, we compared the estimated and predicted head-on-body rotation sensitivities (*Figure 4A*) and phases (*Figure 4B*) for all the Purkinje cells in our sample on a neuron by neuron basis. Overall, estimated and predicted sensitivities and phases were comparable ($R^2$ = 0.77 and 0.75 for sensitivity and phase, respectively, p < 0.001). The similarity between values is shown by the slope of the line fitted to the data which were not different from 1 (p = 0.42 and 0.83 for gain and phase, respectively). Thus, the summation model provided a good estimate of the gain of the preferred direction responses during combined stimulation. Similar results were obtained in our analysis of the non-preferred direction responses (*Figure 4—figure supplement 1*).

As stated above, the generation of vestibulospinal reflexes requires central pathways to explicitly transform vestibular information from a head-centered to a body-centered reference frame during self-motion. To better understand the coding by our population of neurons, we first computed a 'head sensitivity' and a 'body sensitivity' ratio for each neuron (*Figure 5A*, see Materials and methods). Two theoretical neurons, one that selectively encodes head-in-space and the other that selectively encodes body movement, are indicated by red and orange stars, respectively. A neuron selectively encoding head-in-space motion (red star) would display comparable responses to whole-body and head-on-body rotations (head sensitivity ratio = 1), while not responding to body-under-head rotations (body sensitivity ratio = 0). On the other hand, a neuron selectively encoding body motion would display a comparable response to whole-body and body-under-head rotations (orange star, body sensitivity ratio = 1), while not responding to head-on-body rotations (head sensitivity ratio = 0). In contrast, comparison of these ratios across each of the Purkinje cells in our neuronal population revealed considerable heterogeneity in the relationship between these two measures relative to these theoretical neurons. As reviewed above, anterior vermis Purkinje cells target neurons in the deep cerebellar nuclei (i.e., rFN) (*Batton et al., 1977*; *Yamada and Noda, 1987*). Thus, for comparison, we computed these ratios for rFN neurons that had been studied during the same conditions (*Brooks and Cullen, 2009*). The red and orange shaded areas represent the distribution of our sensitivity ratios for the unimodal and bimodal rFN neuron populations reported in this prior study. Notably, in contrast to the Purkinje cells of our present study, the relationship between the rFN unimodal and bimodal neuron sensitivity ratios are well aligned with that of our theoretical neurons that selectively encoded head and body movement, respectively.

To next evaluate the transformation of vestibular information from a head-centered to body-centered reference frame during self-motion for each Purkinje cell, we computed a coding index (see Materials and methods). Specifically, this index compared each neuron's sensitivity when only the head moved relative to space (i.e., head-on-body rotation, *Figure 3*) versus when only the body moved relative to space (i.e., body-under-head rotation, *Figure 2*), to the combined stimulation condition. The results from the analysis of our Purkinje cell population are shown in *Figure 5B*. Indeed, only a minority of neurons were designated as primarily head (26%) or body (2%) encoding (light and dark orange bars, respectively). A complementary analysis of *non-preferred* direction responses (relative to vestibular stimulation) revealed similar results (*Figure 5—figure supplement 1*). Again, the corresponding distribution of coding indices estimated for rFN neurons from *Brooks and Cullen, 2009*, is shown for comparison (*Figure 5B*, top right inset). In contrast to Purkinje cells, the majority of neurons were designated as primarily head (34%) or body (26%) encoding.

## Influence of head position on Purkinje cell's vestibular responses

In theoretical models of reference frame transformations, responses to the sensory inputs are generally modulated by a postural signal (e.g., the position of the head relative to the body) (*Pouget and Snyder, 2000*, *Salinas and Abbott, 2001*). Indeed, neurons in the deep cerebellar nuclei of primates show such tuning. Specifically, the vestibular responses of bimodal neurons in the rFN modulate as a function of head position (*Brooks and Cullen, 2009*; *Kleine et al., 2004*; *Shaikh et al., 2004*). This

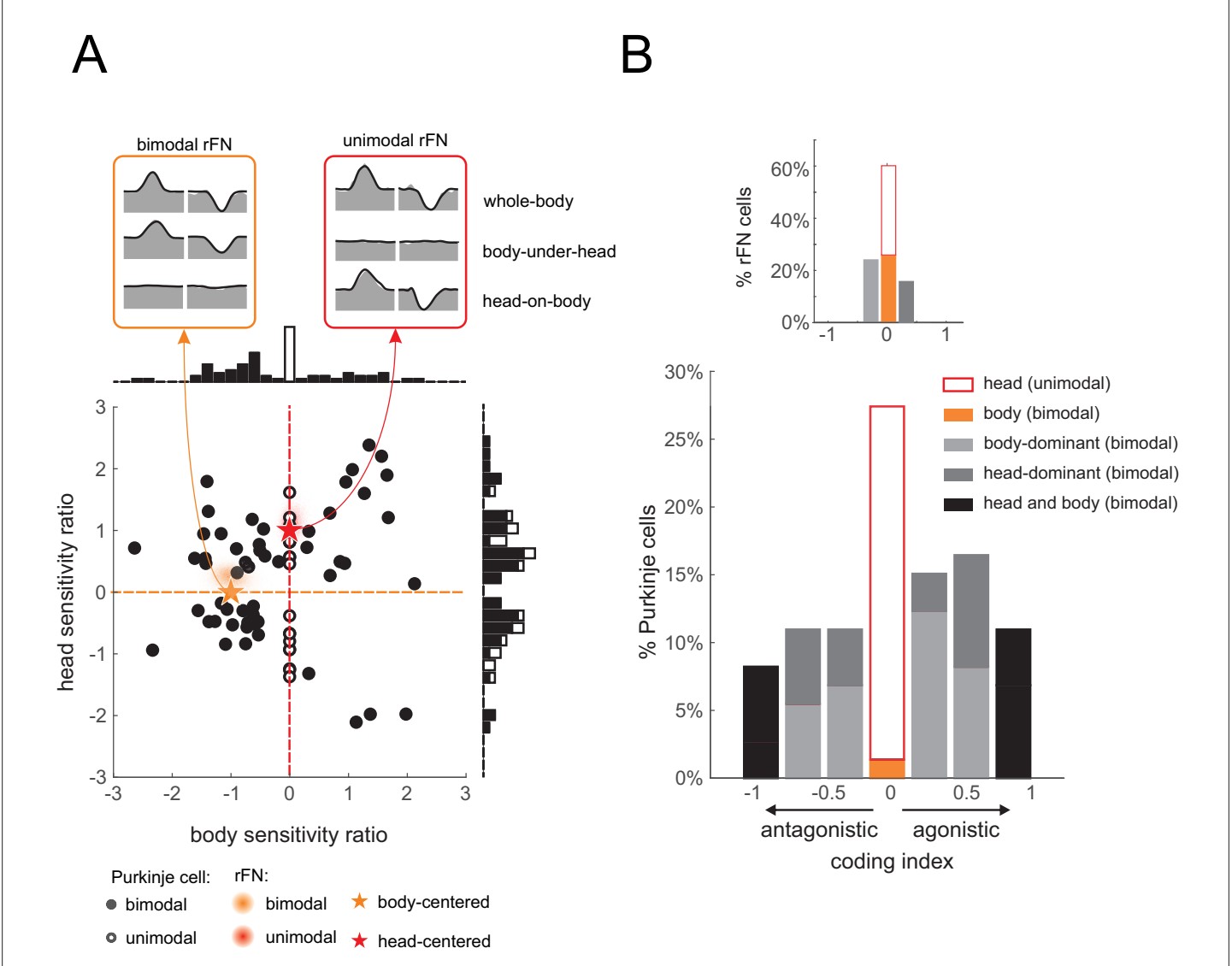

**Figure 5.** Heterogeneity in Purkinje cell simple spike encoding of head and body movement. (**A**) Scatter plot of the relationship between the head sensitivity ratio ($S_{vest.+prop.}/S_{vest}$) and body sensitivity ratio ($S_{prop.}/S_{vest.}$) for the preferred direction. Histograms (top and right) illustrate the distributions of body and head sensitivity ratios, respectively. Orange versus red stars indicate ideal encoding of body versus head movement in space, respectively. For comparison, the red and orange shaded areas representing the distribution of values estimated for unimodal and bimodal rostral fastigial nucleus (rFN) neurons (***Brooks and Cullen, 2009***) are superimposed. *Inset*: examples of the responses of a bimodal (orange) and unimodal (red) rFN neurons during whole-body, body-under-head, and head-on-body movement are shown for comparison (***Figure 5A*** has been adapted from Figure 1 from ***Brooks and Cullen, 2013***). (**B**) Distribution of coding indexes (see Materials and methods). Positive and negative values correspond to agonistic and antagonistic responses to head versus body encoding, respectively. *Inset*: the distribution of coding indices estimated for rFN neurons (***Brooks and Cullen, 2009***) is shown for comparison.

The online version of this article includes the following figure supplement(s) for figure 5:

**Figure supplement 1.** Heterogeneity in Purkinje cell simple spike encoding of head and body movement for the non-preferred direction.

finding has been taken as support for the view that a reference frame transformation of vestibular signals from head- to body-centered occurs in the cerebellar vermis (reviewed in ***Cullen, 2019***). Thus, we next asked: How is this tuning generated? And more specifically, is it computed within the deep cerebellar nuclei or instead inherited from the Purkinje cells that target neurons in the deep cerebellar nuclei? To address these questions, we first determined whether the vestibular responses of Purkinje cells were affected by static changes in head-on-body position (i.e., 'gain-field' condition, see Materials and methods). We measured neuronal responses to vestibular stimulation (i.e., whole-body

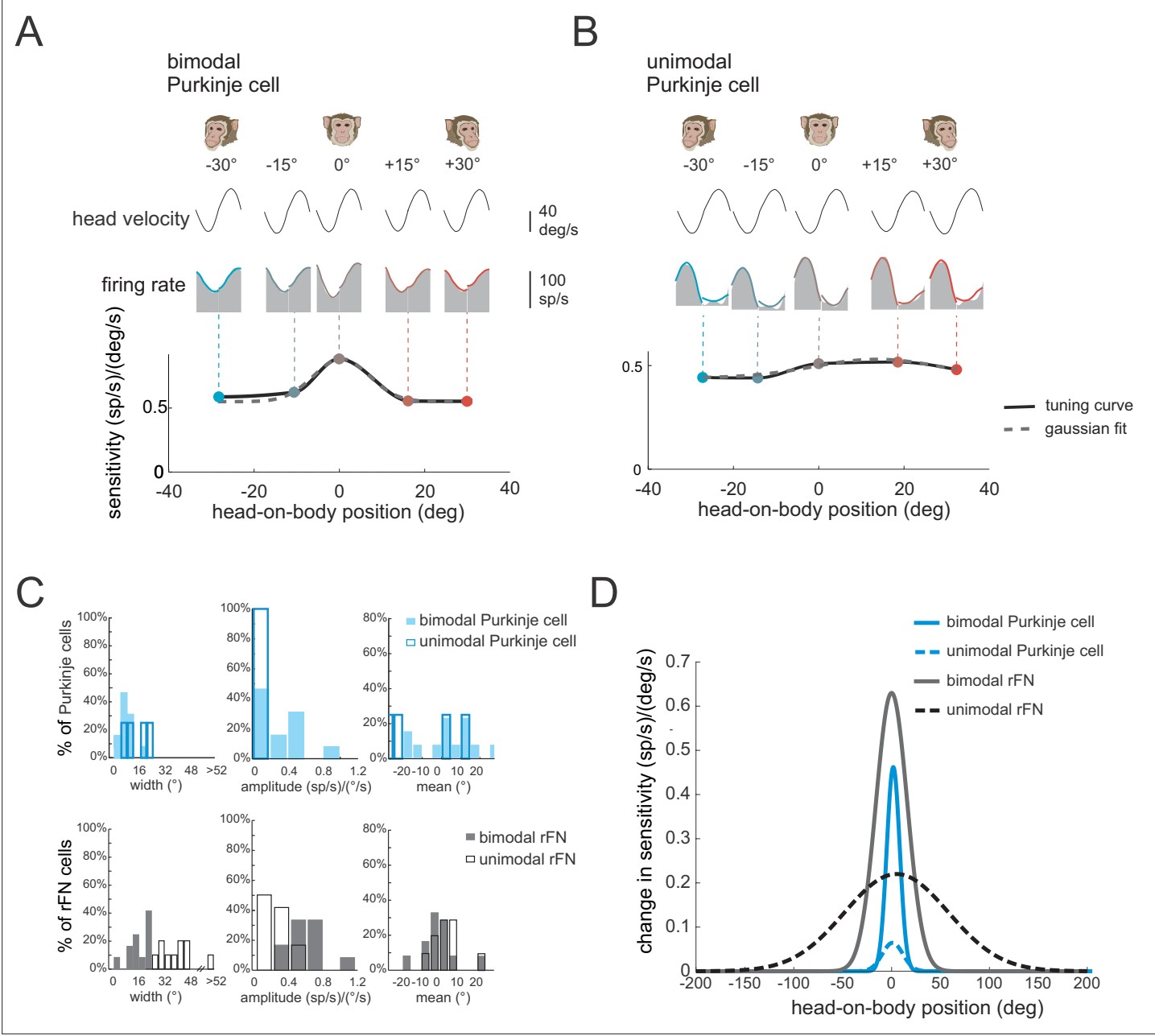

**Figure 6.** The vestibular responses of bimodal Purkinje cells show head-on-body position-dependent tuning. (**A, B**) Tuning curves for the vestibular sensitivities of an example bimodal (**A**) and unimodal (**B**) Purkinje cell measured by applying whole-body rotation with the head oriented at different positions relative to the body. Note, bimodal neurons, but not unimodal neurons, show tuning as a function of head-on-body position. (**C**) *Top panel:* Distributions of tuning widths (left), amplitudes (middle), and means (right) for bimodal (filled bars, N = 13) and unimodal (open bars, N = 4) Purkinje cells. *Bottom panel:* For comparison, the same distributions are plotted for a population of rostral fastigial nucleus (rFN) neurons previously characterized using a comparable approach (***Figure 6C*** has been adapted from Figure 5 from ***Brooks and Cullen, 2009***). (**D**) Average tuning curves computed by aligning the peak of each individual neuron's tuning curve. Average tuning curves are shown for bimodal and unimodal Purkinje cells (blue) for vestibular stimulation with the head oriented at different positions relative to the body. Again, for comparison, the average tuning curves of rFN neurons are superimposed (***Figure 6D*** has been adapted from Figure 6 from ***Brooks and Cullen, 2009***).

rotation) applied with the head positioned at five different orientations ranging from −30 (left) to +30° (right) relative to the body (−30°, −15°, 0°, 15°, and 30°). The example bimodal neuron was typical in that it displayed marked changes in vestibular sensitivity with changes in head-on-body position (***Figure 6A***). In contrast, we did not find evidence for such tuning in unimodal neurons (***Figure 6B***).

To quantify each Purkinje cell's tuning, we fit a Gaussian function to vestibular sensitivity as a function of head position (see Materials and methods), and computed the tuning width, amplitude, and mean direction provided by the best fit to each neuron (*Figure 6C*, top row; filled and open blue bars denote bimodal and unimodal neurons $N = 13$ versus 4, respectively). Note that the small number of unimodal neurons tested in this condition reflects that they constituted a relatively small percentage of our overall Purkinje cell population. First, bimodal neurons were more narrowly tuned than were unimodal neurons (mean tuning widths; 7.2 versus 15°, respectively). Additionally, bimodal neurons showed stronger tuning relative to unimodal neurons (mean tuning amplitude; 0.52 versus 0.05 (sp/s)/ (°/s)), respectively. Finally, there was no difference in the mean of the tuning curve between unimodal neurons and bimodal neurons ($p > 0.37$). We next compared the tuning of our bimodal and unimodal Purkinje cells with that previously described for their target neurons in the rFN (*Brooks and Cullen, 2009*). The corresponding distributions of rFN neuron tuning width, amplitude, and mean direction are plotted in the bottom row of *Figure 6C*. To facilitate comparison between the tuning of Purkinje and rFN cells, we aligned the peak of each individual neuron's tuning curve with zero and averaged the resultant curves across bimodal and unimodal groups for each (*Figure 6D*). Overall, the strength of tuning was significantly higher for bimodal rFN than Purkinje cells (*Figure 6D*, compare solid gray and black lines, ~30% reduction for Purkinje cells, $p < 0.001$). Tuning width was also reduced for bimodal Purkinje cells (~40% reduction), while mean tuning direction was comparable for both cell groups ($p > 0.05$). Moreover, tuning was consistently stronger for bimodal than unimodal neurons in Purkinje cells as has previously been shown for rFN neurons (*Figure 6D*, compare solid and dashed lines). We note that because the interaction between vestibular responses and head-on-body position that underlies the tuning shown in *Figure 6* is inherently nonlinear, this tuning cannot be predicted by the component of the Purkinje cells' dynamic modulation that is in phase with head position during body-under-head rotation (i.e., *Figure 2A*).

## Linear combination of the Purkinje cells' response can encode head and body motion

To summarize, our results have shown that while most vestibular-sensitive Purkinje cells in the anterior vermis integrate vestibular and neck proprioceptive signals, the transformation from head- to body-centered reference frame is not complete. Instead, single bimodal Purkinje cells generally dynamically encoded intermediate representations of self-motion that were between head and body motion. In contrast, bimodal neurons in the deep cerebellar nuclei – the primary target of these Purkinje cells (*Figure 7A*; rFN) – dynamically encode body motion (i.e., orange shaded region, *Figure 5C*) and also show stronger vestibular tuning as a function of head-on-body-position. Thus, taken together, our present results suggest that the transformation from a head- to body-centered representation of self-motion is achieved by integrating the activities of multiple Purkinje cells.

Accordingly, we next tested this hypothesis. Specifically, to quantify the actual number of Purkinje cells necessary to explain the responses of bimodal rFN neurons, we determined whether a simple linear model optimizing the weights of the activities of multiple Purkinje cells (see Materials and methods) could generate bimodal rFN neural responses across conditions (*Figure 7B*). As expected, combining the activities of more Purkinje cells (i.e., increasing population size) led to an increase in the goodness of fit (*Figure 7C*). Datasets used for this modeling first included Purkinje cell responses recorded during (i) our three dynamic conditions (i.e., whole-body, body-under-head, head-on-body rotations) alone (black curve) and (ii) these same three dynamic conditions as well as the the gain-field condition (i.e., *Figure 6*; dashed blue curve). In the latter case, the tuning of Purkinje cells that were not held long enough to test during gain-field paradigm were generated from the tuning curves distribution of the tested neurons (see Materials and methods). Importantly, in both cases we found that the weighted activities of ~40 neurons generated responses that well approximated those previously reported for bimodal rFN neurons (*Figure 7C*, red arrow); the confidence intervals of our model estimations and that of the rFN neural responses completely overlapped for a population of ~40 neurons. Thus, a population of 40 Purkinje cells could explain the dynamic representation of body motion across conditions (*Figure 7D*), as well as robust encoding of vestibular stimuli as a function of static head position observed in bimodal rFN neurons (*Figure 7D*, right panel).

Above we described how Purkinje cells demonstrate considerable heterogeneity in their responses to both vestibular and proprioceptive stimulation. Thus, we next asked whether certain Purkinje cell

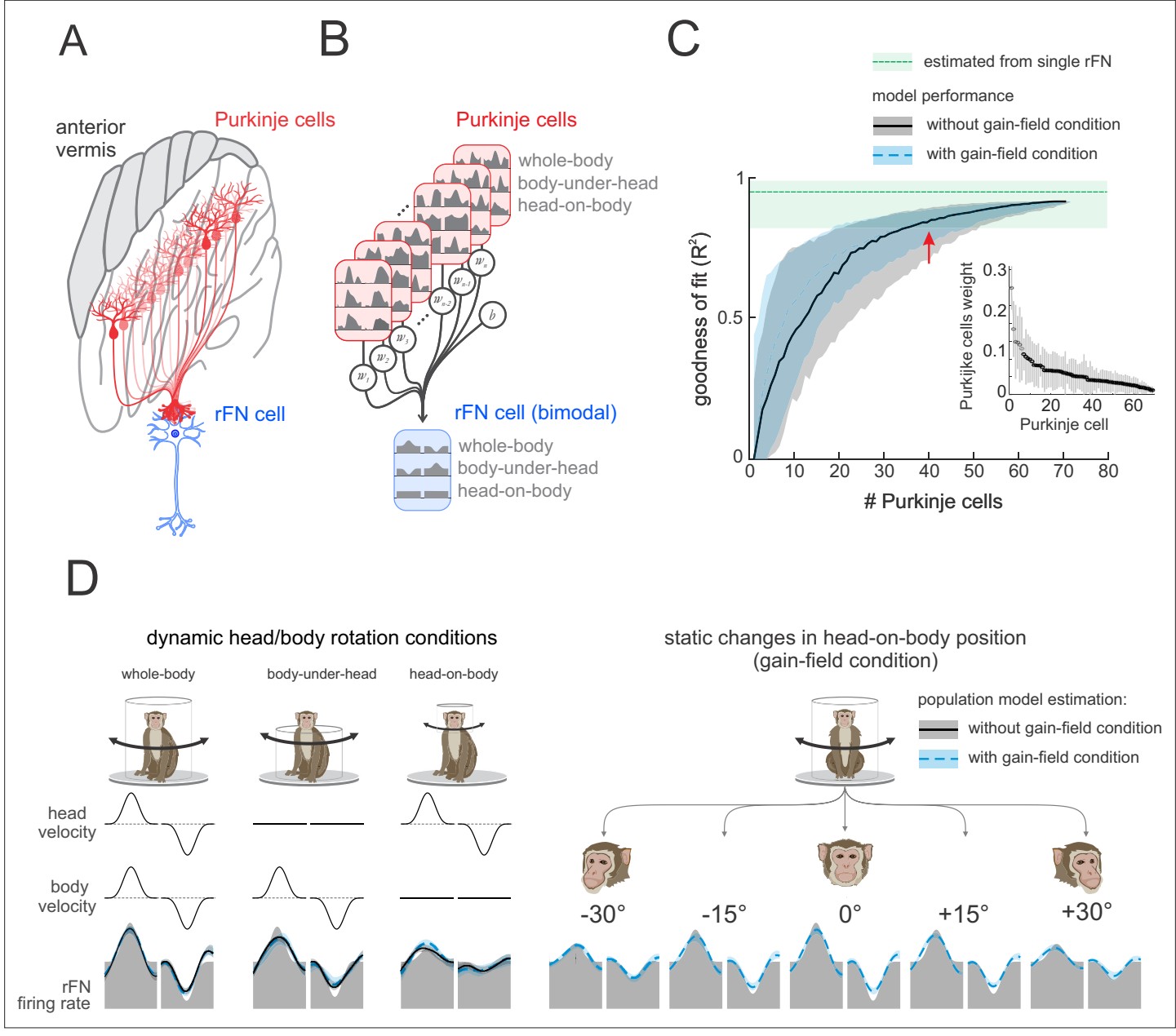

**Figure 7.** A simple linear population model of Purkinje cell integration can explain the responses of target bimodal neurons in deep cerebellar nuclei across all self-movement conditions. (**A**) Illustration of the convergence of multiple Purkinje cells onto a single neuron in the rostral fastigial nucleus (rFN), with different shades of red representing theoretical differences in the weighing of each Purkinje cell's synapse with the target rFN neuron. (**B**) Schematic of the linear summation population model used to estimate the firing rate of a target neuron in the rFN. Each Purkinje cell's weight was optimized to generate the best estimate of the average bimodal rFN neuron across conditions (**Brooks and Cullen, 2009**). (**C**) Model performance as a function of the number of Purkinje cells. Black curve corresponds to model fit to the simple spike firing rates of all 73 Purkinje cells recorded during our three dynamic conditions (i.e., whole-body, body-under-head, and head-on-body movements). Blue curve corresponds to the model fit to simple spike firing rates of all 73 Purkinje cells during these same three dynamic conditions as well as simulated responses of these cells recorded in the gain-field condition (**Figure 6**). The variability estimated from a population of rFN bimodal neurons previously described by **Brooks and Cullen, 2009**, is represented by the green shaded band. *Inset*: the distribution of computed weights for each Purkinje cell modeled during our three dynamic conditions with 40 Purkinje cells, sorted based on average weight. (**D**) Estimated model firing rates based on a population of 40 Purkinje cells superimposed on the actual average firing rate of a bimodal rFN neuron (gray shaded region). Solid black lines versus dashed blue lines illustrate firing rate estimations from models that included (**i**) the three dynamic head/body rotation conditions (left) versus (ii) the three dynamic conditions as well as the gain-field condition (right).

The online version of this article includes the following figure supplement(s) for figure 7:

*Figure 7 continued on next page*

*Figure 7 continued*

**Figure supplement 1.** The distribution of the weights of the inputs to the model with 40 Purkinje cells projecting to a bimodal rostral fastigial nucleus (rFN) neuron.

**Figure supplement 2.** The distribution of the weights of the inputs to a model with 40 Purkinje cells projecting to a unimodal rostral fastigial nucleus (rFN) neuron.

**Figure supplement 3.** Modeling the mossy fiber inputs to unimodal fastigial neurons.

**Figure supplement 4.** Exploring the effect of systematically altering the distribution of gain and phase values in this simulated mossy fiber input.

**Figure supplement 5.** A simple linear population model of Purkinje cell integration can explain the responses of target unimodal neurons in deep cerebellar nuclei across all self-movement conditions.

classes were weighted higher in our population model than others. For example, given that rFN neurons show linear tuning, one might predict that Purkinje cells with linear tuning might be weighted higher in our population model than those with v-shaped tuning. However, we found that this was not the case. The model weight distributions were similar for linear versus v-shaped versus rectifying Purkinje cells. Similarly, model weight distributions were similar for (i) bimodal versus unimodal Purkinje cells, (ii) Type I versus Type II Purkinje cells, as well as (iii) Purkinje cells with agonistic versus antagonistic vestibular and proprioceptive responses. These distributions are illustrated in *Figure 7— figure supplements 1 and 2* for our modeling of bimodal versus unimodal rFN neurons, respectively.

Finally, we note that our simple population model above in *Figure 7A* assumed no input to the rFN neurons other than that from Purkinje cells. However, the fastigial nucleus receives mossy fiber inputs via the vestibular nuclei, reticular formation, and central cervical nucleus (reviewed in *Voogd et al., 1996*) that likely also encode vestibular and/or neck proprioceptive information (e.g., *Roy and Cullen, 2001*, *Kubin et al., 1980*, *Kubin et al., 1981*; *Thomson et al., 1996*). Therefore, we next tested the effect of adding simulated mossy fiber inputs to our model. Prior studies have shown that the dynamics of responses of vestibular nuclei neurons strongly resemble those of unimodal fastigial neurons in rhesus monkeys (i.e., they encode vestibular input and are insensitive to neck proprioceptive inputs, *Roy and Cullen, 2001*). In contrast, the response of neurons in the reticular formation and central cervical nucleus to such yaw head and/or neck rotations have not yet been described. We therefore simulated mossy fiber input first as a summation of vestibular and neck proprioceptive inputs, for which the gains and phases were randomly drawn from a distribution, comparable to that previously reported (*Mitchell et al., 2017*) in the vestibular nuclei (see Materials and methods). We repeated this approach for a total of 1000 simulations (*Figure 7—figure supplement 3*). We then further explored the effect of systematically altering this simulated mossy fiber input relative to the reference distribution of mossy fiber inputs by (i) doubling the gain, (ii) reducing the gain by half, (iii) doubling the phase, and (iv) reducing the phase by half (*Figure 7—figure supplement 4*). Overall, we found that the addition of such simulated mossy fiber inputs did not dramatically alter our estimate of the Purkinje cell population size required to generate rFN neurons responses (~50 versus 40; *Figure 7—figure supplements 3 and 4*). Furthermore, comparable results were obtained for model weight distributions as shown above in *Figure 7—figure supplements 1 and 2*.

Finally, for completeness, we also used the same approach to quantify the number of Purkinje cells necessary to explain the responses of unimodal rFN neurons and obtained comparable results (*Figure 7—figure supplement 5*). Interestingly, our finding that a population of ~40–50 Purkinje cells is again required to explain the responses of bimodal and unimodal rFN neurons matches the value established independently from anatomical studies of Purkinje cell – deep cerebellar nucleus neuron projection ratio. We further consider this point below in the Discussion.

## Discussion
### Summary of results

Here, we recorded the simple spike activity of Purkinje cells of the anterior vermis during passive vestibular (i.e., whole-body rotation), neck proprioceptive (i.e., body-under-head rotation), and a combination of vestibular and neck proprioceptive stimulation (i.e., head-on-body rotation). First, we found that most Purkinje cells responded to both vestibular and neck proprioceptive stimulation (i.e., bimodal neurons). Second, the linear combination of the responses to dynamic neck proprioceptive

and vestibular stimulation alone provided a good estimate of each Purkinje cell's response during combined stimulation. Third, bimodal neurons generally did not encode either the motion of the head or body in space across conditions. Instead, they dynamically encoded intermediate representations of self-motion between head and body motion. Additionally, bimodal neurons, but not unimodal neurons, showed tuning for the encoding of vestibular stimuli as a function of static head position. Finally, using a simple linear population model, we establish that combining inhibitory responses from ~40 to 50 Purkinje cells can explain the responses of target neurons in deep cerebellar nuclei across all self-movement conditions. Thus, our findings in alert monkeys provide new insight into the neural mechanisms underlying the coordinate transformation by which the cerebellum uses neck proprioceptive information to transform vestibular signals from a head- to body-centered reference frame.

## Purkinje cells have diverse temporal responses to dynamic vestibular and neck proprioceptive sensory stimulation

The integration of vestibular and neck proprioceptive-related information is required to convert head-centered vestibular signals to the body-centered reference frame required for postural control (reviewed in *Cullen, 2019*). Our recordings in the anterior vermis of alert monkeys demonstrate that most vestibular-sensitive Purkinje cells also encode neck proprioceptive-related information. As reviewed above, anterior vermis Purkinje cells project to the rFN, the most medial of the deep cerebellar nuclei (*Fujita et al., 2020*; *Husson et al., 2014*) which plays a key role in the control of posture. Two types of rFN neurons have been previously identified in alert monkeys: unimodal and bimodal neurons (*Brooks and Cullen, 2009*). Unimodal neurons respond to vestibular stimulation during passive rotations and dynamically encode head movement. Bimodal rFN neurons respond to both vestibular and neck proprioceptive stimulation and dynamically encode body movement. Notably, because the vestibular and neck proprioceptive sensitivities of rFN bimodal neurons are both equal and complementary in sign, they sum linearly to effectively cancel each other during passive head-on-body rotations – a condition in which both sensory systems are activated but the body does not move in space. In contrast, here, we found that vestibular-sensitive anterior vermis Purkinje cells were on average more sensitive to vestibular than proprioceptive stimulation and that there was considerable variability in the relative signs of responses to each modality. As a result, cancellation of these two inputs during passive head-on-body rotations was the exception rather than the rule (i.e., *Figure 3B*) with the vast majority of bimodal Purkinje cells demonstrating significant modulation in response to passively applied head-on-body rotations. Thus, unlike bimodal rFN neurons, which dynamically encode body motion, bimodal Purkinje cells dynamically encode an intermediate representation of self-motion.

## Reference frame transformations: Purkinje cell's vestibular responses modulated by posture

Theoretical models of reference frame transformations commonly include a sensory input (e.g., vestibular or visual information) that is modulated by a postural signal (e.g., head-on-body position) (*Pouget and Snyder, 2000*; *Salinas and Abbott, 2001*). The resultant modulation of the sensory signal is commonly referred to as a gain field (*Andersen and Mountcastle, 1983*) and is thought to be mediated via nonlinear interactions between sensory responses and head/body referenced cues (*Zipser and Andersen, 1988*; *Salinas and Abbott, 2001*). Our present results reveal the neural substrate of such a reference frame transformation required for postural control. Notably, bimodal anterior vermis Purkinje cells displayed vestibular tuning as a function of head-on-body position during horizontal rotations. This tuning is similar but not as strong as that shown by downstream bimodal neurons in the target rFN (*Brooks and Cullen, 2009*), and indeed some rFN neurons do encode vestibular information in a body-centered reference frame for both two-dimensional (*Kleine et al., 2004*; *Shaikh et al., 2004*) and three-dimensional (*Martin et al., 2018*) self-motion. Thus, our present data establish that the modulation of vestibular information by a postural signal becomes more marked in the progression from the cerebellar cortex to the deep cerebellar nuclei. Interestingly, in the present study, such nonlinear interactions between neck position and vestibular signals were only in bimodal and not unimodal vestibular Purkinje cells. Future experiments are required to understand the implications of the dynamic coding of head rather than body movements by the unimodal neurons.

Finally, it is noteworthy that our findings regarding the transformation from a vestibular to head-centered reference frame in the anterior vermis of the alert primate contrast with those of Pompeiano, Manzoni, and colleagues in anesthetized decerebrate cats. Using dynamic tilt stimuli, they concluded that the direction of average response vector of Purkinje cells encoding both vestibular and proprioceptive information well corresponded to body tilt – consistent with a complete transformation from head- to body-centered reference frame (*Denoth et al., 1979*; *Manzoni et al., 1998*; *Manzoni et al., 2004*). One potential explanation for this apparent difference in neural strategy is that our studies were performed in intact alert behaving animals whereas Manzoni and colleagues completed their experiments in anesthetized decerebrate preparation where modulation/gating by cortical structures is not present. Additionally, there are significant differences across species regarding how the vestibular system integrates multimodal information even at the first stage of central processing in the vestibular nuclei (reviewed in *Cullen, 2019*). For instance, vestibular nuclei neurons in alert mice, cats, and cynomolgus monkeys commonly display vestibular-proprioceptive convergence (*Medrea and Cullen, 2013*; *Cullen, 2016*; *McCall et al., 2017*). In contrast, in rhesus monkeys, vestibular nuclei neurons are only sensitive to vestibular input, and instead, proprioceptive information is integrated only at the subsequent levels of vestibular processing, most notably in the deep nuclei of the cerebellum (*Roy and Cullen, 2001*; *Brooks and Cullen, 2009*; *Carriot et al., 2013*).

## Population coding: the heterogeneous response of Purkinje cells and convergence in the rFN

Our results establish that there is considerable heterogeneity in the response dynamics of anterior vermis Purkinje cells to vestibular and/or neck proprioceptive sensory stimulation. Semicircular canal afferents and vestibular nuclei neurons provide the primary source of vestibular information to the cerebellum via mossy fiber input. However, while they encode head velocity with a phase lead, the responses of individual Purkinje cells actually more often lagged rather than led head velocity. Albus and Marr proposed that the divergent feedforward mossy fiber projections onto a far larger number of granule cells effectively expand the dimensionality of neural space, in turn allowing better downstream decoding to linearly classify dynamic patterns of activity (*Marr, 1969*; *Albus, 1971*). Indeed, recent studies have shown that mossy fibers from multiple sensory systems converge on each of more than 50 billion granule cells (*Chabrol et al., 2015*; *Knogler et al., 2017*; *Lanore et al., 2021*), with interneurons likely further contributing to the temporal diversity of granule cell responses (*Rousseau et al., 2012*; *Kennedy et al., 2014*). In turn, >100,000 granule cells project to a single Purkinje cell via parallel fibers (*Fujishima et al., 2018*). Thus, together these features of the cerebellar microcircuitry are well designed to generate high-dimensional dynamic coding of information by Purkinje cells relative to their mossy fiber input across regions of the cerebellum including the anterior vermis.

In this context, the heterogeneity we observed in anterior vermis Purkinje cells' responses then contrasts strikingly with the responses of their target neurons in rFN (*Brooks and Cullen, 2009*). Our estimation that pooling the responses of a population of ~40–50 Purkinje cells can explain more homogeneous responses of bimodal and unimodal rFN neurons matches that established independently from anatomical studies of the Purkinje cell – deep cerebellar nucleus neuron projection ratio in rodents and cats (*Person and Raman, 2012*; *Palkovits et al., 1977*). Additionally, these Purkinje cells likely send direct projections to the vestibular nuclei. Testing with comparable stimulation protocols has established that the responses of vestibular nuclei neurons are comparable to those of unimodal rFN neurons (compare *Cullen, 2019*, with *Brooks and Cullen, 2014*). Thus, our present modeling results regarding the population convergence required to account for unimodal rFN neurons can be directly applied to vestibular nuclei neurons. Interestingly, Purkinje cells can display patterns of neuronal synchrony during active movements (*Person and Raman, 2012*; *Sarnaik and Raman, 2018*; *Wu and Raman, 2017*) which could, in turn, alter the timing and modulation of target neuron responses in the deep cerebellar nuclei in a nonlinear manner. Nevertheless, we found that responses could be predicted using a simple linearly weighted summation of ~40–50 neurons. In this context, we note our modeling was based on averaged Purkinje cell and rFN neuron responses. Future studies including simultaneous recordings from Purkinje cells and rFN neurons can provide additional insight into whether comparable population sizes can account for single trial responses in real time.

Finally, it is noteworthy that our study focused on the sensory responses of Purkinje cells (i.e., responses to vestibular and/or proprioceptive stimulation) during *passively* applied self-motion.

Prior studies focused on the responses of Purkinje cells during *voluntary* movements have similarly concluded that they are more heterogeneous than those of their target neurons in the deep cerebellar nuclei (e.g., saccades: *Thier et al., 2000*; wrist control: *Tomatsu et al., 2016*). Interestingly, in their analysis, *Tanaka et al., 2019*, likewise estimated that linearly pooling the responses of ~40–50 Purkinje cells could account for the more homogeneous responses of target neurons in the deep cerebellar nucleus (i.e., the dentate nucleus) during voluntary wrist movements. We speculate that expanded dimensionality of the cerebellum provides a basis set for sensorimotor errors as well as plasticity at the level of Purkinje cells required to generate accurate movements (reviewed in *Sohn et al., 2021*) as well as ensure robust calibration over time. Overall, our current results reveal a striking transformation from heterogeneous response dynamics of cerebellar Purkinje cells to more stereotyped response dynamics of neurons in the targeted deep cerebellar nucleus. These findings provide new insights into the neural computations that ultimately ensure accurate postural control in our daily lives.

## Materials and methods

### Experimental model and subject details

Animal experimentation: All experimental protocols were approved by the Johns Hopkins University Animal Care and Use Committee and were in compliance with the guidelines of the United States National Institutes of Health (PR19M408). The cerebellar recordings were conducted in two male macaque monkeys (*Macaca mulatta*). The animals were housed on a 12 hr light/dark cycle. The recording sessions were about three times a week, for approximately 2 hr each session. Both animals had participated in previous studies in our laboratory, but they were in good health condition and did not require any medication.

### Method details

#### Surgical procedures

The two animals were prepared for chronic extracellular recording using aseptic surgical techniques described previously (*Massot et al., 2012*). Briefly, animals were pre-anesthetized with ketamine hydrochloride (15 mg/kg i.m.) and injected with buprenorphine (0.01 mg/kg i.m.) and diazepam (1 mg/kg i.m.) to provide analgesia and muscle relaxation, respectively. Loading doses of dexamethasone (1 mg/kg i.m.) and cefazolin (50 mg/kg i.v.) were administered to minimize swelling and prevent infection, respectively. Anticholinergic glycopyrrolate (0.005 mg/kg i.m.) was also preoperatively injected to stabilize heart rate and to reduce salivation, and then again, every 2.5–3 hr during surgery. During surgery, anesthesia was maintained using isoflurane gas (0.8–1.5%), combined with a minimum 3 l/min (dose adjusted to effect) of 100% oxygen. Heart rate, blood pressure, respiration, and body temperature were monitored throughout the procedure. During the surgical procedure, a stainless-steel post for head immobilization and recording chambers were fastened to each animal's skull with stainless-steel screws and dental acrylic. Craniotomy was performed within the recording chamber to allow electrode access to the cerebellar cortex. An 18-mm-diameter eye coil (three loops of Teflon-coated stainless-steel wire) was implanted in one eye behind the conjunctiva. Following surgery, we continued dexamethasone (0.5 mg/kg i.m.; for 4 days), anafen (2 mg/kg day 1, 1 mg/kg on subsequent days), and buprenorphine (0.01 mg/kg i.m.; every 12 hr for 2–5 days, depending on the animal's pain level). In addition, cefazolin (25 mg/kg) was injected twice daily for 10 days. Animals recovered in 2 weeks before any experimenting began.

#### Data acquisition

During the experiments, the monkey sat in a primate chair secured to a turntable, and its head was centered in a coil system (CNC Engineering). Extracellular single-unit activity was recorded using enamel-insulated tungsten microelectrodes (Frederick-Haer). The location of the anterior vermis of the cerebellar cortex was determined relative to the abducens nucleus identified based on stereotypical neuronal responses during eye movements. The Purkinje cells were identified by their characteristic complex spike activity. The angular velocity of the turntable was measured using a gyroscope sensor (Watson Industries, Eau Claire, WI). Monkeys' gaze and head angular positions were measured using the magnetic search coil technique. The neck torque produced by the monkey against its head restraint was measured using a reaction torque transducer (QWFK-8M; Honeywell, Canton, MA). All

analog behavioral signals were low-pass filtered with a 125 Hz cut-off frequency and acquired at 1 kHz. The neural activity was recorded at 30 kHz using a data acquisition system (Blackrock Microsystems). Action potentials from the neural recording were sorted using a custom Matlab GUI (MathWorks), which provides threshold, clustering, and manual selection/removal methods.

## Head and body motion paradigms

Two monkeys were trained to follow a target projected onto a cylindrical screen located 60 cm away from the monkey's head. Each neuron's insensitivity to saccades and ocular fixation was confirmed by having the head-restrained monkey attend to a target that stepped between horizontal positions over a range of ±30°. Each neuron's lack of response to eye movements was further confirmed by absent responses to smooth pursuit eye movements during sinusoidal target motion (0.5 Hz, 40°/s peak velocity).

Next, to characterize each Purkinje cell's vestibular and proprioceptive sensitivities, we applied rotational stimuli mimicking the monkey's head movement generated during ±30° orienting gaze shifts in the head-unrestrained condition (i.e., 'active-like' condition). Use of this head movement trajectory facilitates direct comparison with rFN and vestibular nuclei neurons (e.g., *Brooks and Cullen, 2014*). First, vestibular sensitivities were assessed by applying whole-body rotations about an earth-vertical axis in the dark (i.e., whole-body-rotations). Second, neck proprioceptive sensitivities were assessed by rotating the monkey's body with this same active-like trajectory while its head was held stationary relative to space (i.e., body-under-head rotations). Third, neural sensitivities to combined propriocep-tive and vestibular stimulation were assessed by passively rotating the monkey's head relative to its stationary body (i.e., head-on-body rotations) with this same trajectory. Finally, in a subset of neurons, we also applied whole-body rotations about an earth-vertical axis in the dark (1 Hz, ±40°/s) with the head statically oriented at five different positions relative to the body (−30°, −15°, 0°, 15°, and 30°) to assess whether static neck position influenced vestibular-induced modulation during whole-body sinusoidal rotation (i.e., the 'gain-field' condition).

Histological analysis confirmed that the Purkinje cells were located in lobules II–V of the anterior vermis, ~0–2 mm from the midline. We note that while we first tested the vestibular sensitivity of individual neurons, we did also test whether neurons that were insensitive to vestibular stimulation responded to neck proprioceptive stimulation. Consistent with Manzoni and colleagues' prior studies in anesthetized cat (12%, *Manzoni et al., 1998*), we found that only a small portion of Purkinje cells (~10%) fell into this latter category.

## Data analysis

Analysis of neuronal discharge dynamics: Data were imported into the Matlab (MathWorks) program-ming environment for analysis, filtering, and processing as previously described (*Dale and Cullen, 2019*). Neuronal firing rate was computed by filtering spike trains with a Kaiser window at twice the frequency range of the stimulus (*Cherif et al., 2008*). We first verified that each neuron neither paused nor burst during saccades and was unresponsive to changes in eye position during fixation. We then used a least-squares regression analysis to describe each Purkinje cell simple spike's response to whole-body and body-under-head rotations:

$$\hat{fr}(t) = b + c_{p,i}X_i(t) + c_{v,i}\dot{X}_i(t) + c_{a,i}\ddot{X}_i(t) \tag{1}$$

where $fr(t)$ is the estimated firing rate, $b$ is a bias term, $c_{p,i}$, $c_{v,i}$, and $c_{a,i}$ are coefficients repre-senting the position, velocity, and acceleration sensitivities respectively to head ($i = 1$) or body motion ($i = 2$), and $X_i$, $\dot{X}_i$, and $\ddot{X}_i$ are head ($i = 1$) or body ($i = 2$) position, velocity, and acceleration (during whole-body and body-under-head rotations), respectively. This least-squares regression was solved for non-negative and non-positive criterion to ensure sign consistency across estimated coefficients. For each model coefficient in the analysis, we computed 95% confidence intervals using a nonpara-metric bootstrap approach ($n = 2000$; *Carpenter and Bithell, 2000*; *Sylvestre and Cullen, 1999*). All non-significant coefficients were set to zero. We then used coefficients to estimate the sensitivity and phase of the response using the following equations:

$$Sensitivity = sgn\left(c_{p,i}, c_{p,i}, c_{p,i}\right) \times \sqrt{\frac{\left((2\pi f)^2 c_{a,i} - c_{p,i}\right)^2 + (2\pi f c_{v,i})^2}{(2\pi f)^2}} \tag{2}$$

$$Phase = tan^{-1}\left(\frac{(2\pi f)^2 c_{a,i} - c_{p,i}}{2\pi f c_{v,i}}\right) \tag{3}$$

For which $f = 1$Hz to match the duration of half-cycle of movements (500 ms) and the sign term (i.e., $sgn\left(c_{p,i}, c_{p,i}, c_{p,i}\right)$) equals either 1 or –1 for positive versus negative coefficients, respectively. The sensitivity of the Purkinje cells to the neck proprioceptive stimulation (during body-under-head rotations) was used to categorize the cells into unimodal (zero sensitivity) and bimodal (non-zero sensitivity).

Neuronal tuning to vestibular and proprioceptive inputs was further categorized as linear, rectifying, or V-shaped. Linear neurons demonstrated increased and decreased firing rates in the preferred and non-preferred directions, respectively. The difference between the magnitude of sensitivities in each of the two directions was within 0.2 (sp/s)/(°/s). Rectifying neurons demonstrated increased firing rate in the preferred direction and minimal modulation (i.e., sensitivity smaller than 0.2 (sp/s)/(°/s)) in the non-preferred direction. V-shaped neurons demonstrated an increased firing rate in both directions. The difference between the magnitude of their sensitivities in each of the two directions was within 0.2 (sp/s)/(°/s). Finally, neurons that did not fit any of these criteria were characterized as 'other'. Note that v-shaped neurons were categorized as Type I or II based on the direction for which their vestibular sensitivity was larger, since their responses in each direction were not identical.

We used a similar approach to estimate sensitivities to passive head-on-body movements. Since in this condition, it is not possible to dissociate neck proprioceptive and vestibular sensitivities, we estimated them as a single coefficient. Estimated sensitivities were compared to those predicted from the linear summation of the vestibular and proprioceptive sensitivities estimated for the same neuron during passive whole-body and body-under-head rotations (termed summation model), respectively. To quantify the ability of the linear regression analysis to model neuronal discharges, the variance-accounted-for (VAF) for each regression equation was determined as previously described (**Cullen et al., 1996**). Values are expressed as mean ± SD and paired-sample Student's t-tests were used to assess differences between conditions.

Quantifying head versus body encoding: We computed a 'head sensitivity ratio' and 'body sensitivity ratio' for each Purkinje cell. These ratios were defined as the neuron's (i) sensitivity to head-on-body rotation/sensitivity to whole-body rotation and (ii) sensitivity to body-under-head rotation/sensitivity to whole-body rotation, respectively. Further to quantify the relative encoding of head versus body motion by a given cell, we computed a 'coding index', which was defined as the ratio (smaller value)/(larger value) of these two ratios.

Quantification of head position on Purkinje cell vestibular sensitivity: The tuning curves for different head-on-body positions were fit with Gaussian curves with the following equation:

$$S = Ae^{\frac{-\left(H_{position} - \mu\right)^2}{2\sigma^2}} \tag{4}$$

where $\mu$ represents the mean, $\sigma$ is a measure of the width, and $A$ is the amplitude from the peak to the base of the Gaussian curve (as described previously, **Brooks and Cullen, 2013**).

Population modeling of Purkinje cells: To determine whether integrating the activities of multiple Purkinje cells could explain the response of their target neurons in the rFN, we used the linear model below:

$$r\hat{F}N = \sum_{i=1}^{N} w_i \times Pcell_i \tag{5}$$

where $r\hat{F}N$ is a reconstructed firing rate response of an rFN neuron. The $w_i$ corresponds to weights of connection from Purkinje cells to an rFN neuron, which all considered non-positive to reflect inhibitory synapses from Purkinje cells to rFN neurons. $Pcell_i$ are observed firing rate of simple spikes $N$ Purkinje cells, where $N$ is a number between 1 and the total number of Purkinje cells in the dataset. For each $N$, we used a bootstrapping approach to find the 95% confidence intervals of the goodness of fit ($R^2$ as well as the model predictions).

To model the population response of Purkinje cells during the 'gain-field condition', we first fit a Gaussian function of the tuning curve of 13 bimodal Purkinje cells that were recorded during this

condition. Next, we used the parameters of these Gaussian functions to find a normal distribution representing the tuning curves of bimodal Purkinje cells. Then, for the remaining Purkinje cells that were not recorded during the 'gain-field' condition, we generated tuning curves by drawing from this normal distribution. Since four unimodal Purkinje cells that were recorded during the 'gain-field' condition did not demonstrate significant tuning, we did not consider any tuning to the remaining unimodal Purkinje cells.

Finally, we modeled the contribution of the mossy fiber input to the rFN as a summation of independent responses to vestibular and neck proprioceptive stimulation. To simulate the mossy fiber input, we randomly selected response gains and phases from normal distributions that described the responses of neurons in the vestibular nuclei (i.e., 0.6 ± 0.1 (sp/s)/(°/s) and 20 ± 5°, respectively), and repeated this for a total of 1000 simulations. We further assessed the robustness of our modeling performance by modifying these distributions. Specifically, we tested the effect of either doubling or halving the gain (i.e., 1.2 ± 0.2 and 0.3 ± 0.05 (sp/s)/(°/s), respectively) or phase (i.e., 40 ± 10° and 10 ± 2.5°, respectively), resulting simulations based on four modifications of the original distribution.

## Acknowledgements

This research was supported by grants R01-DC002390 and R01-DC018061 from the National Institutes of Health (KEC). We would like to thank Dale Roberts for his technical support and Jessica Brooks, Robyn Mildren, Pum Wiboonsaksakul, and members of the Cullen lab for helpful discussions.

## Additional information

### Funding

| Funder | Grant reference number | Author |
| --- | --- | --- |
| National Institute on Deafness and Other Communication Disorders | R01-DC002390 | Kathleen E Cullen |
| National Institute on Deafness and Other Communication Disorders | R01-DC018061 | Kathleen E Cullen |

The funders had no role in study design, data collection and interpretation, or the decision to submit the work for publication.

### Author contributions

Omid A Zobeiri, Data curation, Formal analysis, Investigation, Validation, Visualization, Writing – original draft, Writing – review and editing; Kathleen E Cullen, Conceptualization, Funding acquisition, Investigation, Supervision, Writing – original draft, Writing – review and editing

### Author ORCIDs

Omid A Zobeiri ⬤ http://orcid.org/0000-0002-1813-1354
Kathleen E Cullen ⬤ http://orcid.org/0000-0002-9348-0933

### Ethics

All experimental protocols were approved by the Johns Hopkins University Animal Care and Use Committee and were in compliance with the guidelines of the United States National Institutes of Health.(PR19M408).

### Decision letter and Author response

Decision letter https://doi.org/10.7554/eLife.75018.sa1
Author response https://doi.org/10.7554/eLife.75018.sa2

## Additional files

### Supplementary files
• MDAR checklist

### Data availability
All data and codes to generate figures are available on Figshare under the URL: https://doi.org/10.6084/m9.figshare.19362239.

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
