## [Editor Report]

This paper addresses the important question of how the cerebellum transforms multiple streams of sensory information into an estimate of the motion of the body in the world. The authors find that Purkinje cells, the inhibitory principal neurons of the cerebellar cortex, have multimodal and highly diverse responses to vestibular and neck proprioceptive inputs. Notably, this information is combined in a way that is different from what is seen in downstream fastigial neurons, which reflect either head or body motion, but not both.

---

## [Decision Letter]

**Decision letter after peer review:**

Thank you for submitting your article "Distinct representations of body and head motion are dynamically encoded by Purkinje cell populations in the macaque cerebellum" for consideration by *eLife*. Your article has been reviewed by 3 peer reviewers, and the evaluation has been overseen by a Reviewing Editor and Andrew King as the Senior Editor. The following individual involved in review of your submission has agreed to reveal their identity: Stephen H Scott (Reviewer #2).

Essential revisions:

1) All Reviewers had concerns about the rFN model, in particular (a) the absence of mossy fiber inputs to fastigial neurons, and (b) whether it's just easiest to get this fastigial response by down-weighting the PCs with ipsilaterally-signed vestibular and proprioceptive signals, which would make it less interesting. More information about the model is required. At a minimum, please add (1) additional discussion of the assumptions and limitations of the model, and (2) analysis of the distribution of model weights over different classes of PCs (especially linear vs v-shaped or rectifying). Please also see the more detailed comments from the individual reviews below.

2) There was concern during the consultation phase that the focus of the paper is the difference in sensitivity to vestibular vs proprioceptive stimuli, and yet different stimuli are being presented in the two conditions (e.g., compare body velocity in Figure 1A Cell 1, and Figure 2A, Cell 1). It was not clear how these differences might impact the regression coefficients and model performance. The velocity differences between conditions should be addressed directly in the manuscript. Specifically, the reason for the discrepancy (and, if relevant, the experimental constraints on matching the velocities) should be explained in the Methods and discussed explicitly in the text. The reviewers further agreed that the authors should address this issue by (1) providing the total variance-accounted-for for all neurons (not just the normalized values in Supp 1and2), (2) plotting the residuals vs predicted firing rate for all neurons, and (3) providing additional justification that the model is not overfit, if possible.

*Reviewer #1 (Recommendations for the authors):*

– The writing, particularly in the Results section, is very abrupt. The reader has to go to the Methods section immediately to understand what stimulus is even being provided, let alone what the motivation for the experiments is. Even as someone with reasonable experience in the vestibular field, I had difficulty following the logic of the experimental design.

– The authors have used a "naturalistic" rotational stimulus, about which I have several questions. First, I didn't understand the motivation for using this. Second, based on the figures, this rotational stimulus varied significantly across experiments. E.g., the head and velocity traces in Figure 1A are quite different for Cell 1, Cell 2, and Cell 3. Why deliver different stimuli if the goal is to make comparisons across cells? Even more confusing, the body velocity traces delivered to those exact same cells (Figure 2) appear again quite different. Why deliver different body velocity stimuli in the proprioceptive versus vestibular experiments? Since the entire manuscript focuses on the comparison between these stimuli, it would seem important to hold the parameters constant across the experiments. Head velocity traces are also different, or appear different, across the rotations at different head-on-body angles (Figure 5; e.g. the head velocity stimulus in 5A for +15 vs +30 is different in amplitude).

– In Figure 5, the authors make the interesting observation that vestibular sensitivity of PCs actually varies depending on the head angle relative to the body. I was expecting that they would then test whether the neck proprioceptive "position" term (that they had derived from the experiments in Figure 2) could be used to predict this shift in sensitivity. Instead, the authors directly jump to comparisons with the rostral fastigial population recorded in a prior study. Can they calculate whether their proprioceptive position weights (c_a,2 from equation 1) explain the measured difference in vestibular sensitivity?

– In Figure 6, the authors show modeling evidence that linear summation of Purkinje cells can produce the tuning results of fastigial neurons. They don't address whether an alternate explanation, that some of the proprioceptive or vestibular tuning derives from direct mossy fiber afferents to fastigial neurons, is possible. Furthermore, they don't seem to address the basic surprise here, that although the PCs show much less "cancellation" between vestibular and proprioceptive inputs (p. 11, last paragraph) than fastigial neurons do, the summation of PC inputs can produce fastigial responses. One assumes that the only way to obtain this result is preferential summation of the small set of PCs that do show cancellation (oppositely signed proprioceptive and vestibular sensitivities) but this is never discussed. If that is the underlying explanation of their model's success, then it seems quite weak.

*Reviewer #2 (Recommendations for the authors):*

I think the first experiment is strong and the paper should emphasize even more the comparison between Purkinje cells and the downstream fastigial neurons. I think some example fastigial neurons could be added to contrast with the Purkinje cell responses. Importantly, Figure 4C shows a red and orange cloud to denote the rFN neurons, but this is very hard to see in the figure. I think the individual neurons should be plotted to highlight the substantive difference between these Purkinje cells and fastigial neurons.

I'm not sure the model is that insightful given that it assumes that there is no other input to the fastigial nucleus. I assume that it would have substantive vestibular and neck proprioceptive input from mossy fibers. As stated in the comment above, is the point of the Purkinje cell activity to counter the sensory input from the mossy fibers in order to create two distinct signals, one related to head and one to body motion in the fastigial population? Thus, a variant for the model would be to show how the population model could create distinct head and body motion signals to counter the presence of a random pattern of vestibular and neck proprioceptive input (assumed to be generated by mossy fiber input). A more advanced model might be interesting in which weights of the parallel fibers onto Purkinje neurons are altered given random mossy fiber inputs to granule cells and fastigial neurons leading to distinct head and body motion signals in the fastigial neurons, but this is certainly beyond the scope of this manuscript.

The cells in Figure 1 and 2 show the same patterns (linear, v-shaped, rectified) across the two conditions. I assume this was common for bimodal cells? Was there any analyses done to support or refute this?

Figure 1 shows that all neurons were sensitive to vestibular input, and no unimodal proprioceptive neurons. Is this an order effect of the experiment in that you always tested vestibular first, and if responsive, then you completed the rest of the experiments? Thus, could there be proprioceptive only neurons, but they were not examined in this study due to the protocol. This is important and needs to be clearly stated in the manuscript, one way or another.

The small sample size makes conclusions regarding head-on-body position a bit of a challenge. Notably, there are only 4 unimodal neurons for Purkinje cells in Figure 5C. The number of neurons is only noted in the figure legend and the number of neurons for the fastigial nucleus is not stated at all and looks to be 11 for bimodal and 10 for unimodal. I think the main text needs to clearly state the actual numbers and recognize the very small sample size for this analysis. These limited number of neurons to characterize tuning properties also weakens the modelling results and conclusions about 40 neurons necessary to predict fastigial neuron responses.

In Figure 4C it looks like the positive body sensitivity ratio neurons tend to have larger values than the negative sensitivity ratio neurons. Is this just a random sampling issue?

*Reviewer #3 (Recommendations for the authors):*

1 – "All non-significant coefficients were set to zero." In the Methods, the authors should briefly describe the rationale for using this approach instead of L1 regularization.

2 – The acceleration term appears to dominate in the non-preferred direction for vestibular stimulation (Figure S1A); why?

3 – Typo: "Albus and Marrs" -> "Albus and Marr".

---

## [Author Response]

Essential revisions:1) All Reviewers had concerns about the rFN model, in particular (a) the absence of mossy fiber inputs to fastigial neurons, and (b) whether it's just easiest to get this fastigial response by down-weighting the PCs with ipsilaterally-signed vestibular and proprioceptive signals, which would make it less interesting. More information about the model is required. At a minimum, please add (1) additional discussion of the assumptions and limitations of the model, and (2) analysis of the distribution of model weights over different classes of PCs (especially linear vs v-shaped or rectifying). Please also see the more detailed comments from the individual reviews below.

We thank the reviewers for their attention to our manuscript and for raising these important points. We have address each of the reviewers’ main concerns.

First, to consider the influence of mossy fiber inputs, we followed Reviewer #2’s suggestion and modeled mossy fiber input using random patterns of vestibular and neck proprioceptive input. Prior studies have shown that the dynamics of vestibular nuclei neuron responses strongly resemble those of unimodal fastigial neurons in rhesus monkeys (i.e., they encode vestibular input and are insensitive to neck proprioceptive inputs, Roy and Cullen, 2001). In contrast, reticular formation neurons responses to such yaw head and/or neck rotations have not yet been described. We therefore simulated mossy fiber input first as a summation of vestibular and neck proprioceptive inputs, for which the gains and phases were randomly drawn from a distribution, comparable to that previously reported (Mitchell er al. 2017) in the vestibular nuclei (Figure 7—figure supplement 3). We then further explored the effect of systematically altering this simulated mossy fiber input – relative to the reference distribution of mossy fiber inputs – by i) doubling the gain, ii) reducing the gain by half, iii) doubling the phase, and iv) reducing the phase by half (Figure 7—figure supplement 4). Overall, we found that the addition of such simulated mossy fiber did not dramatically alter our estimate of the population Purkinje cell population size required to generate rFN neurons responses (~50 versus 40; Figure 7—figure supplement 3 and 4).

Second, to address the reviewers’ concerns regarding the Purkinje cell weights, we have added a new inset to Figure 7C. As can be seen, model weights are well distributed across different Purkinje cells. Further, to confirm that the distribution of the weights of Purkinje cells inputs are distributed over different classes of PCs we now illustrate the weight distributions for (a) linear vs. v-shaped vs. rectifying Purkinje cells, (b) bimodal vs. unimodal Purkinje cells, (c) Type I vs. Type II Purkinje cells and (d) Purkinje cells with agonistic vs. antagonistic vestibular and proprioceptive sensitivities. These results are shown in Figure 7-supplemental figures 1 and 2.

2) There was concern during the consultation phase that the focus of the paper is the difference in sensitivity to vestibular vs proprioceptive stimuli, and yet different stimuli are being presented in the two conditions (e.g., compare body velocity in Figure 1A Cell 1, and Figure 2A, Cell 1). It was not clear how these differences might impact the regression coefficients and model performance. The velocity differences between conditions should be addressed directly in the manuscript. Specifically, the reason for the discrepancy (and, if relevant, the experimental constraints on matching the velocities) should be explained in the Methods and discussed explicitly in the text. The reviewers further agreed that the authors should address this issue by (1) providing the total variance-accounted-for for all neurons (not just the normalized values in Supp 1and2), (2) plotting the residuals vs predicted firing rate for all neurons, and (3) providing additional justification that the model is not overfit, if possible.

We thank the reviewers for bringing this to our attention. The apparent discrepancy was due to an error in finalizing Figure 2. The left panel of Figure 2A illustrated a different cell than that shown in Figures1A and 3B, plotted at different time/amplitude scales than indicated by the bars. We have corrected this figure in the revised manuscript. Indeed, movement trajectories were well matched across all of our conditions, and we have revised the METHODS to more clearly explain how velocities were matched across conditions. Additionally, as requested, we now present the total variance-accounted-for for all neuron responses during whole-body and body-under-head stimulation in Figure 1—figure supplement 1 and Figure 2—figure supplement 2, respectively. Finally, we have also revised the METHODS to explain our linear regression method more clearly. Briefly, as stated in our response to Reviewer #3, a bootstrapping approach was used to prevent overfitting instead of lasso L1 regularization, since we used a non-negative/non-positive criterion in the least-square linear regression and to the best of our knowledge, there is no commonly used package for non-negative lasso regularization.

Reviewer #1 (Recommendations for the authors):– The writing, particularly in the Results section, is very abrupt. The reader has to go to the Methods section immediately to understand what stimulus is even being provided, let alone what the motivation for the experiments is. Even as someone with reasonable experience in the vestibular field, I had difficulty following the logic of the experimental design.

We appreciate the reviewer’s concern and have revised the Results section to more clearly introduce both the stimuli that were applied, as well as the logic that motivates each experiment.

– The authors have used a "naturalistic" rotational stimulus, about which I have several questions. First, I didn't understand the motivation for using this. Second, based on the figures, this rotational stimulus varied significantly across experiments. E.g., the head and velocity traces in Figure 1A are quite different for Cell 1, Cell 2, and Cell 3. Why deliver different stimuli if the goal is to make comparisons across cells? Even more confusing, the body velocity traces delivered to those exact same cells (Figure 2) appear again quite different. Why deliver different body velocity stimuli in the proprioceptive versus vestibular experiments? Since the entire manuscript focuses on the comparison between these stimuli, it would seem important to hold the parameters constant across the experiments. Head velocity traces are also different, or appear different, across the rotations at different head-on-body angles (Figure 5; e.g. the head velocity stimulus in 5A for +15 vs +30 is different in amplitude).

We have revised the Methods and Results sections to more clearly introduce the stimuli that were applied. Specifically, we chose these rotational stimuli for 2 main reasons: (1) First they mimic typical head movements generated by monkeys during orienting gaze shifts in the head-unrestrained condition. Thus, use of these stimuli facilitates future comparisons with neuronal responses during active head rotations. (2) Second, comparable stimuli have been used to assess the responses of both rFN and vestibular nuclei neurons to applied vestibular, proprioceptive, and combined stimulation in rhesus monkeys (see for example Brooks et al., 2015). Thus, the use of these stimuli also facilitates the direct comparison between rFN and vestibular nuclei neurons and the Purkinje cells in the present study. To improve clarity, we have revised the text and now refer to these stimuli as “active-like” motion profiles in the revised METHODS, consistent with the terminology used by Brooks et al., 2015.

We also agree that it is important to apply comparable stimuli in all three paradigms (whole-body, body-under-head, and head-on-body). Indeed, this was the case in our study. As explained in the response to the reviewers’ consensus comments the apparent inconsistency across the example cells in old Figure 2 was due to a scaling error that has now been corrected. We apologize for this oversight. A similar issue has also been corrected in figure 5A, and we have confirmed that the scaling is now correct for all figures.

– In Figure 5, the authors make the interesting observation that vestibular sensitivity of PCs actually varies depending on the head angle relative to the body. I was expecting that they would then test whether the neck proprioceptive "position" term (that they had derived from the experiments in Figure 2) could be used to predict this shift in sensitivity. Instead, the authors directly jump to comparisons with the rostral fastigial population recorded in a prior study. Can they calculate whether their proprioceptive position weights (c_a,2 from equation 1) explain the measured difference in vestibular sensitivity?

We thank the reviewer for this suggestion. We note, however, that it is not theoretically possible to predict a neuron’s “gain field” tuning via the linear addition its response to the static head-on-body position, which was estimated during body-under-head rotation (i.e., Figure 2A). This is because the estimated position term is a fixed value which would not contribute to a head position-dependent change in dynamic firing rate modulation. Put another way, the interaction between vestibular responses and head-on-body position that underlies the tuning shown in Figure 6 (old Figure 5) is inherently nonlinear. We have revised the RESULTS to discuss this point.

– In Figure 6, the authors show modeling evidence that linear summation of Purkinje cells can produce the tuning results of fastigial neurons. They don't address whether an alternate explanation, that some of the proprioceptive or vestibular tuning derives from direct mossy fiber afferents to fastigial neurons, is possible. Furthermore, they don't seem to address the basic surprise here, that although the PCs show much less "cancellation" between vestibular and proprioceptive inputs (p. 11, last paragraph) than fastigial neurons do, the summation of PC inputs can produce fastigial responses. One assumes that the only way to obtain this result is preferential summation of the small set of PCs that do show cancellation (oppositely signed proprioceptive and vestibular sensitivities) but this is never discussed. If that is the underlying explanation of their model's success, then it seems quite weak.

As noted in our response to the reviewers’ consensus feedback above, we have performed additional modeling to consider the influence of mossy fiber inputs and present these new results at the end of the revised RESULTS. Briefly, following Reviewer #2’s suggestion, we modeled mossy fiber input using random patterns of vestibular and neck proprioceptive input drawn from the experimentally distribution observed in the vestibular nuclei. As shown in Figure 7-supplemental figure 3, our new analysis demonstrates that any of these simulated mossy fiber inputs can be combined with Purkinje cell inputs to create two distinct signals, one related to head and one to body motion – consistent with the responses of target unimodal and bimodal rFN neurons. respectively. We then further explored the effect of systematically altering this simulated mossy fiber input – relative to the reference distribution of mossy fiber inputs – by i) doubling the gain, ii) reducing the gain by half, iii) doubling the phase, and iv) reducing the phase by half (Figure 7—figure supplement 4). Overall, we found that the addition of such simulated mossy fiber did not dramatically alter our estimate of the population Purkinje cell population size required to generate rFN neurons responses (~50 versus 40; Figure 7—figure supplement 3 and 4).

Finally, regarding the comment that “the PCs show much less "cancellation" between vestibular and proprioceptive inputs”, we believe that the reviewer is referring to the fact that because vestibular vs. proprioceptive sensitives are typically of opposite sign for bimodal rFN neurons they effectively cancel during combined stimulation (i.e., head-on-body), whereas this is not observed for Purkinje cells. We agree that it is interesting to understanding the relative contribution of agonistic vs. antagonistic Purkinje cells. Thus, in the revised manuscript we now report the weight distributions of Purkinje cells with agonistic vs. antagonistic vestibular and proprioceptive sensitivities. These results are shown in Figure 7-supplemental figures 1and2. Notably, our new findings confirm that both classes have similar weights, such that the heterogeneity that we observed Purkinje cell’s response is represented in the input to the model of fastigial responses.

Reviewer #2 (Recommendations for the authors):I think the first experiment is strong and the paper should emphasize even more the comparison between Purkinje cells and the downstream fastigial neurons. I think some example fastigial neurons could be added to contrast with the Purkinje cell responses. Importantly, Figure 4C shows a red and orange cloud to denote the rFN neurons, but this is very hard to see in the figure. I think the individual neurons should be plotted to highlight the substantive difference between these Purkinje cells and fastigial neurons.

We thank the reviewer for this suggestion. We have revised Figure 5A (old Figure 4C) to include plots of example rFN neurons from our previous study (i.e., Brooks and Cullen 2009) to emphasize the difference between Purkinje cells and fastigial neurons responses across paradigms.

I'm not sure the model is that insightful given that it assumes that there is no other input to the fastigial nucleus. I assume that it would have substantive vestibular and neck proprioceptive input from mossy fibers. As stated in the comment above, is the point of the Purkinje cell activity to counter the sensory input from the mossy fibers in order to create two distinct signals, one related to head and one to body motion in the fastigial population? Thus, a variant for the model would be to show how the population model could create distinct head and body motion signals to counter the presence of a random pattern of vestibular and neck proprioceptive input (assumed to be generated by mossy fiber input). A more advanced model might be interesting in which weights of the parallel fibers onto Purkinje neurons are altered given random mossy fiber inputs to granule cells and fastigial neurons leading to distinct head and body motion signals in the fastigial neurons, but this is certainly beyond the scope of this manuscript.

As noted above, we have performed additional modeling to directly address the reviewer’s concern. These new results are presented at the end of the revised RESULTS. Briefly, following the reviewer’s suggestion, we modeled mossy fiber input using random patterns of vestibular and neck proprioceptive input drawn from the experimentally distribution observed in the vestibular nuclei. As shown in Figure 7-supplemental figure 3, our new analysis demonstrates that any of these simulated mossy fiber inputs can be combined with Purkinje cell inputs to create two distinct signals, one related to head and one to body motion – consistent with the responses of target unimodal and bimodal rFN neurons. respectively. We then further explored the effect of systematically altering this simulated mossy fiber input – relative to the reference distribution of mossy fiber inputs – by i) doubling the gain, ii) reducing the gain by half, iii) doubling the phase, and iv) reducing the phase by half (Figure 7—figure supplement 4). Overall, we found that the addition of such simulated mossy fiber did not dramatically alter our estimate of the population Purkinje cell population size required to generate rFN neurons responses (~50 versus 40; Figure 7—figure supplement 3 and 4).

The cells in Figure 1 and 2 show the same patterns (linear, v-shaped, rectified) across the two conditions. I assume this was common for bimodal cells? Was there any analyses done to support or refute this?

We thank the reviewer for this observation and suggestion. We performed an additional analysis (Figure 2—figure supplement 3) and found that the category (i.e., linear, v-shaped, rectified, etc.) of most of the Purkinje cells is generally not the same across the two conditions. Thus, Purkinje cells show considerable heterogeneity in their simple spike responses dynamics to vestibular versus proprioceptive stimulation. We now note this in the RESULTS.

Figure 1 shows that all neurons were sensitive to vestibular input, and no unimodal proprioceptive neurons. Is this an order effect of the experiment in that you always tested vestibular first, and if responsive, then you completed the rest of the experiments? Thus, could there be proprioceptive only neurons, but they were not examined in this study due to the protocol. This is important and needs to be clearly stated in the manuscript, one way or another.

The reviewer is correct that we generally first tested the vestibular sensitivity of individual neurons. We however did check whether neurons that were insensitive to vestibular stimulation responded to neck proprioceptive stimulation. We found that only a minority of Purkinje cells (~10%) fell into this latter category – a result is similar to that of Manzoni and colleagues in anesthetized cat (12%, Manzoni et al., 1998). We now provide this information in the revised Methods.

The small sample size makes conclusions regarding head-on-body position a bit of a challenge. Notably, there are only 4 unimodal neurons for Purkinje cells in Figure 5C. The number of neurons is only noted in the figure legend and the number of neurons for the fastigial nucleus is not stated at all and looks to be 11 for bimodal and 10 for unimodal. I think the main text needs to clearly state the actual numbers and recognize the very small sample size for this analysis. These limited number of neurons to characterize tuning properties also weakens the modelling results and conclusions about 40 neurons necessary to predict fastigial neuron responses.

As noted above, we have revised the Results to clarify the numbers of Purkinje cells that were tested (13 bimodal and 4 unimodal Purkinje cells). For comparison, in our Brooks and Cullen study, tuning curves were computed for 10 bimodal and 12 unimodal rFN. We note that i) unimodal Purkinje cells make up a relatively small percentage of anterior vermis Purkinje cells and ii) similar to unimodal rFN, our small sample of unimodal Purkinje cells did not demonstrate significant tuning. In contrast, all bimodal Purkinje cells in our sample demonstrated significant tuning. To simulate responses for the bimodal Purkinje cells that were not held long enough to test during gain-field paradigm (i.e., Figure 6), we generated tuning curves drawn from a normal distribution estimated from 13 bimodal Purkinje cells. We appreciate this was not clear in the original submission and have revised the Methods section to clarify our approach. Overall, while we recognize that our sample size is small, we nevertheless found it interesting that including this our results from this protocol did not increase the estimated population size relative to that estimated using our other dynamic protocols.

In Figure 4C it looks like the positive body sensitivity ratio neurons tend to have larger values than the negative sensitivity ratio neurons. Is this just a random sampling issue?

We thank the reviewer for this comment and performed additional analyses, but did not find difference between the amplitude of negative vs. positive values of body sensitivity ratios (-1.0 ± 0.5 vs. 1.1 ± 0.6; p=0.39).

Reviewer #3 (Recommendations for the authors):1 – "All non-significant coefficients were set to zero." In the Methods, the authors should briefly describe the rationale for using this approach instead of L1 regularization.

A non-negative/non-positive least-square linear regression used to assign a sign to each neuron’s response sensitivity. A bootstrapping approach was used instead of lasso L1 regularization as a non-parametric to eliminate the effect of nonsignificant coefficients, since to the best of our knowledge there is no commonly used package for non-negative lasso regularization (see revised Methods).

2 – The acceleration term appears to dominate in the non-preferred direction for vestibular stimulation (Figure S1A); why?

We agree that this is an interesting feature of the non-preferred response. It indicates that Purkinje cell responses generally led stimulation velocity. While our studies do not address why this is the case, we now emphasize this point in the revised caption in reference to the relevant Figure (Figure 1-supplementary figure 1).

3 – Typo: "Albus and Marrs" -> "Albus and Marr".

Thank you, we have corrected this typo.